# Towards Predicting Any Human Trajectory In Context

**Ryo Fujii**[1,2]    **Hideo Saito**[1,2]    **Ryo Hachiuma**[3]
[1]Keio University    [2]Keio AI Research Center    [3]NVIDIA

## Abstract

Predicting accurate future trajectories of pedestrians is essential for autonomous systems but remains a challenging task due to the need for adaptability in different environments and domains. A common approach involves collecting scenario-specific data and performing fine-tuning via backpropagation. However, the need to fine-tune for each new scenario is often impractical for deployment on edge devices. To address this challenge, we introduce TrajICL, an In-Context Learning (ICL) framework for pedestrian trajectory prediction that enables adaptation without fine-tuning on the scenario-specific data at inference time without requiring weight updates. We propose a spatio-temporal similarity-based example selection (STES) method that selects relevant examples from previously observed trajectories within the same scene by identifying similar motion patterns at corresponding locations. To further refine this selection, we introduce prediction-guided example selection (PG-ES), which selects examples based on both the past trajectory and the predicted future trajectory, rather than relying solely on the past trajectory. This approach allows the model to account for long-term dynamics when selecting examples. Finally, instead of relying on small real-world datasets with limited scenario diversity, we train our model on a large-scale synthetic dataset to enhance its prediction ability by leveraging in-context examples. Extensive experiments demonstrate that TrajICL achieves remarkable adaptation across both in-domain and cross-domain scenarios, outperforming even fine-tuned approaches across multiple public benchmarks. *Project Page*.

## 1 Introduction

Predicting pedestrian trajectories is crucial for applications such as autonomous driving [11], robot navigation [26, 17], and surveillance systems [13]. Recent learning-based methods have shown strong performance in trajectory prediction [1, 23, 55, 70, 28, 41, 42, 22, 43, 56, 19, 18, 59]. However, most existing approaches are trained and evaluated within specific environments or domains, limiting their generalizability. For real-world deployment, autonomous systems must handle diverse scenarios, requiring trajectory prediction models to be robust across varying environments and domains (*e.g.*, map layouts, camera positions, and sensor types). The lack of adaptability to these factors significantly hinders their practical applicability. A common solution involves collecting scenario-specific data and fine-tuning models [60, 35, 18]. However, this approach requires on-device backpropagation for adaptation using the collected data at the target environment, which is often impractical on edge devices due to the high computational and memory costs associated with training. Furthermore, managing multiple models tailored to different scenarios increases system complexity. Therefore, developing a single, adaptable model capable of generalizing across diverse real-world environments without fine-tuning remains an open challenge.

To address this challenge, we explore the use of In-Context Learning (ICL) [7, 68, 54] for trajectory prediction (Figure 1). ICL enables models to adapt to new tasks using only a few demonstration examples provided as part of the input, without requiring updates to the model weights. Unlike fine-tuning, which modifies model weights through backpropagation, ICL operates solely via forward

39th Conference on Neural Information Processing Systems (NeurIPS 2025).

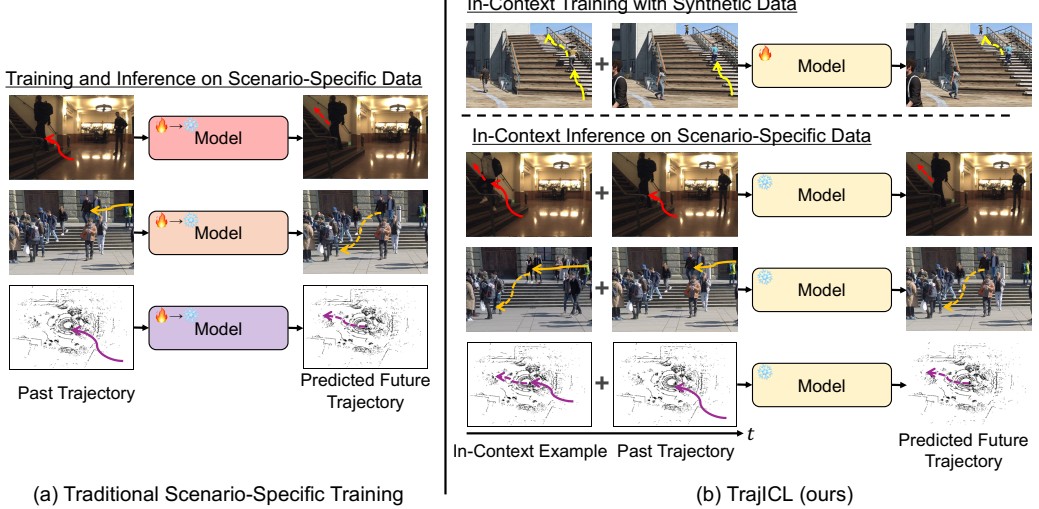

Figure 1: Illustration of real-world trajectory prediction scenarios and the adaptation pipeline. (a) The adaptation pipeline of traditional methods, where models are trained on scenario-specific data. (b) The adaptation pipeline of our proposed TrajICL, which automatically selects examples and adapts to novel scenarios by leveraging the scenario-specific examples without requiring training on scenario-specific data.

passes, keeping the model weights fixed. This key difference means that adaptation occurs during inference without requiring a computationally intensive training phase or storing gradients, making it highly suitable for resource-constrained edge devices. Leveraging this capability, we utilize a small set of examples to enable a single model to quickly adapt to diverse scenarios. This approach facilitates the reuse of trajectory models across a wide range of scenarios, eliminating the need for costly fine-tuning.

However, incorporating ICL into trajectory prediction presents three major challenges: (1) Emerging ICL capability for trajectory prediction: Randomly selected examples provide minimal ICL capability to the model. Even when examples are selected from the same scene, if they represent different spatial locations or exhibit divergent movements, the effectiveness in improving the model's adaptation is limited. Such variations fail to provide the model with sufficiently relevant context to generalize effectively to new scenarios. (2) Suboptimal selection of examples based on past trajectory input: Existing methods select examples based on similarity to the input query (*i.e.*, past trajectory). However, short past trajectory segments often fail to capture long-term intentions. Moreover, pedestrian motion is inherently multi-modal, where similar past trajectories can lead to divergent future trajectories due to subtle influencing factors. Consequently, relying solely on past trajectories for the example selection, without accounting for long-term dynamics, can result in misleading examples and hinder effective in-context adaptation. (3) Adaptability to diverse scenarios: Existing real-world datasets [81, 49, 30, 53, 5, 75, 45] have limited scenario diversity, often focusing on specific environments and interaction patterns. Training on these small-scale datasets restricts ICL's ability to generalize to unseen scenarios, reducing the model's adaptability to a broader range of real-world situations.

To address these challenges, we propose **TrajICL**, an ICL framework for trajectory prediction. First, we introduce a spatio-temporal similarity-based example selection (STES) that identifies relevant examples exhibiting similar motions at comparable locations in the past. By training the model using these spatially and temporally similar examples selected through STES, we enable the development of ICL capability, allowing the model to effectively adapt to new scenarios with minimal examples and without the need for additional training. Second, to address the suboptimal selection based on past trajectory, we propose a prediction-guided example selection (PG-ES) that refines the example selection process using prediction results. PG-ES consists of two phases. First, the model predicts the future trajectory based on its own past trajectory and examples selected according to their similarity with the query past trajectory using the proposed STES. In the second phase, the predicted future trajectory, in addition to the past trajectory, is utilized for example selection. This refinement process

allows the model to select more relevant examples by incorporating long-term dynamics. Finally, instead of relying on small real-world datasets with limited scenario diversity, we train the model on a large-scale synthetic dataset [14] to learn predictive capabilities using in-context examples. This allows the trained model to be directly applicable to edge devices in various environments, enhancing its adaptability to real-world conditions.

Our main contributions are as follows: 1) We propose an ICL framework for trajectory prediction, enabling adaptation to diverse scenarios without training on the scenario-specific data. Our approach leverages large-scale synthetic datasets to enhance generalization to real-world conditions, overcoming the limitations of small-scale real-world datasets to learn predictive capabilities using in-context examples. 2) We introduce a spatio-temporal similarity-based example selection (STES) that allows the model to acquire ICL capability. 3) We present prediction-guided example selection (PG-ES), which utilizes both past and predicted future trajectories to select more relevant examples, rather than relying solely on past trajectory input.

## 2   Related Work

**Adapting Trajectory Prediction.** Pedestrian trajectory prediction aims to predict future positions based on past trajectories. Deep learning methods demonstrate strong performance due to their representational capabilities [1, 23, 55, 70, 28, 41, 42, 22, 43, 56, 19, 18]. Despite the significant advancements, previous trajectory prediction models are often tailored to specific training domains, limiting their generalizability to new scenarios. To address this, recent research has explored lightweight adaptation techniques for pretrained models. Some approaches focus on cross-domain transfer [73, 25, 35], while others emphasize online adaptation [48, 33, 10], continual learning [44, 69, 58], or prompt tuning-based strategies [60]. While these methods enhance forecasting performance, they often incur high computational costs due to backpropagation on the scenario-specific samples and require multiple models for different scenarios. In contrast, we leverage a small number of examples to seamlessly adapt to diverse scenarios, eliminating the need for costly fine-tuning.

**In-Context Learning.** In-Context Learning (ICL) [7, 68, 54] enables large language models (LLMs) to perform new tasks by providing a few input-output examples, or demonstrations, alongside the task input. ICL offers several advantages over traditional model adaptation methods, which typically involve pre-training followed by fine-tuning. A key benefit of ICL is that it circumvents the need for fine-tuning, which can be limited by computational resource constraints. Compared to parameter-efficient fine-tuning (PEFT) methods [24, 4], ICL is computationally cheaper and preserves the model's generality by leaving the model parameters unchanged. Following its success in Natural Language Processing (NLP), ICL has been extended to various domains within Computer Vision (CV), including images [4, 63, 64, 80, 3, 50, 76, 62, 31], video [27], point clouds [16, 57], and skeleton sequences [65], showcasing its ability to generalize across diverse, unseen tasks. Recent studies [29, 38] have explored the use of LLMs in vehicle trajectory prediction, highlighting their effectiveness in in-context learning [38]. In contrast, our approach prioritizes equipping lighter trajectory prediction models with in-context learning capability, offering greater efficiency and practicality for real-world applications.

**Prompt Selection.** The effectiveness of In-Context Learning (ICL) is highly influenced by the selection of relevant examples [37, 47, 32]. Previous studies have shown that selecting in-context examples that are closer to the query improves performance [37, 67, 77]. To address this challenge, several methods have been proposed to select examples that are more similar to the query for ICL [37, 51, 78]. A simple approach is to retrieve the nearest neighbors of the input query based on their similarities [37, 51]. To enhance the robustness of ICL, some studies have employed iterative methods [51, 27, 31, 36]. However, in trajectory prediction tasks, a typical example selection based solely on past trajectory input can be suboptimal. In this work, we introduce a prediction-guided prompt selection approach that selects in-context examples based on their similarity to both the input past trajectory and the predicted future trajectory, aiming to choose more effective examples considering long-term dynamics.

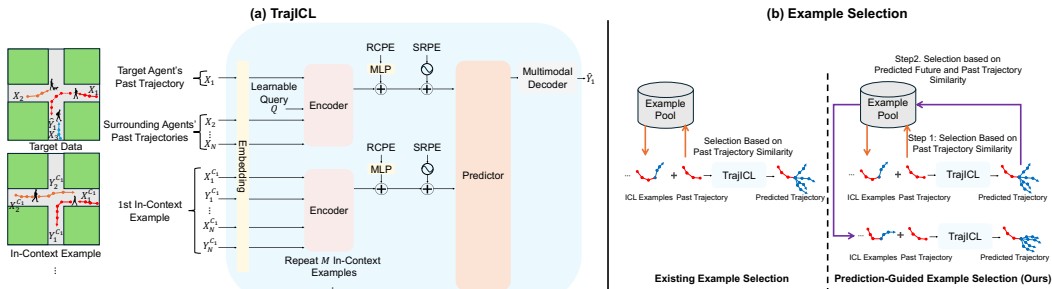

Figure 2: An illustration of our TrajICL framework. (a) The overall architecture includes an embedding layer, a trajectory encoder, an in-context-aware trajectory predictor, and a multi-modal decoder. (b) Rather than relying solely on past trajectories for the example selection, we introduce prediction-guided example selection, which leverages both past and predicted future trajectories to identify more relevant examples.

# 3 TrajICL

## 3.1 Problem Formulation

**Trajectory Prediction with In-Context Learning.** Trajectory prediction aims to forecast the future positions of a target agent based on its own past trajectory and the trajectories of surrounding agents. Formally, let $X = (X_1, X_2, \ldots, X_N) \in \mathbb{R}^{N \times T_{\text{obs}} \times 2}$ denote the past trajectories of $N$ pedestrians over $T_{\text{obs}}$ time steps. The trajectory of the $n$-th pedestrian is defined as $X_n = (x_1^n, x_2^n, \ldots, x_{T_{\text{obs}}}^n) \in \mathbb{R}^{T_{\text{obs}} \times 2}$, where each position $x_t^n = (u_t^n, v_t^n) \in \mathbb{R}^2$ indicates the location at time $t$. We consider the primary $X_1$ as the target pedestrian. Let $Y_1 = (Y_1^1, Y_1^2, \ldots, Y_1^K)$ denote the set of $K$ possible future trajectories of the target pedestrian over $T_{\text{pred}}$ time steps, where each predicted trajectory is $Y_1^k = (y_1^1, y_2^1, \ldots, y_{T_{\text{pred}}}^1) \in \mathbb{R}^{T_{\text{pred}} \times 2}$, and $y_t^1 = (u_t^1, v_t^1) \in \mathbb{R}^2$. Both $X$ and $Y$ are preprocessed such that the last observed position of the primary agent at time step $T_{\text{obs}}$ is shifted to the origin. The goal is to learn a mapping function $\mathcal{F}$ from the past trajectories $X$ to the set of future trajectories $Y_1$, such that $Y_1 = \mathcal{F}(X)$.

In this study, we propose leveraging in-context learning to enable trajectory prediction models to adapt to diverse scenarios without the need for updating model parameters through backpropagation. To predict $Y_1$, we use the past trajectories $X$ in conjunction with an in-context set $\mathcal{C}$: $Y_1 = \mathcal{F}(X, \mathcal{C})$, where $\mathcal{C} = (\tilde{X}^i, \tilde{Y}_1^i)_{i=1}^M$ contains $M$ pairs of past trajectories and the primary agent's ground-truth future trajectory. These example pairs are selected from the example pool, $\mathcal{D}^s$, which consists of pairs of past trajectories and their corresponding ground-truth future trajectories, all coming from the same scene $s$ as the one to which $X$ belongs.

## 3.2 Spatio-temporal Similarity-based Example Selection (STES)

While random selection of in-context learning examples represents the most straightforward approach, our experiments empirically demonstrate that this method leads to suboptimal performance for trajectory prediction tasks. To address this limitation, we introduce a novel approach called spatio-temporal similarity-based example selection (STES) that automatically identifies the most relevant examples for enhanced trajectory prediction.

The core insight driving our proposed STES method is that trajectories with similar past patterns are likely to exhibit similar future behaviors. Building on this intuition, we propose retrieving the top-$M$ past trajectories along with their corresponding ground-truth future trajectories based on a carefully designed similarity metric. Formally, given a past trajectory query $X_1$ and an example pool $D^s$, we identify the most relevant in-context examples as:

$$\mathcal{C} = \text{Top-}M_{\tilde{X}_1^i \in \mathcal{D}^s} \left[ S(X_1, \tilde{X}_1^i) \right], \quad |\mathcal{C}| = M, \tag{1}$$

where $\mathcal{C}$ represents the set of top-$M$ relevant trajectories selected from the example pool $\mathcal{D}^s$ according to the similarity metric $S(X_1, \tilde{X}_1^i)$. This metric quantifies the similarity between the target agent's past trajectory and each candidate primary agent's past trajectory in the example pool.

In our introduced STES approach, we define spatio-temporal similarity as $S(X_1, \tilde{X}_1^i) = \sigma(S_p(X_1, \tilde{X}_1^i)) + \sigma(S_v(X_1, \tilde{X}_1^i))$. $S_p$ and $S_v$ are the spatial and temporal similarities defined as follows:

$$S_p(X_1, \tilde{X}_1^i) = \frac{1}{1 + d_p(X_1, \tilde{X}_1^i)}, \quad S_v(X_1, \tilde{X}_1^i) = \frac{1}{1 + d_v(X_1, \tilde{X}_1^i)}, \tag{2}$$

where $d_p(\cdot, \cdot)$ and $d_v(\cdot, \cdot)$ represent the mean squared errors (MSE) of position and velocity, respectively, capturing both spatial proximity and motion similarity between trajectories. $\sigma(\cdot)$ is the normalization function that scales the similarity value within the range $[-1, 1]$ via min-max normalization across the entire set of similarities between the target agent's past trajectory and all candidate trajectories in the example pool.

### 3.3 Prediction-Guided Example Selection (PG-ES)

To address the suboptimal selection based solely on past trajectory, we propose prediction-guided example selection (PG-ES), which utilizes the trajectory prediction results to refine the example selection process. PG-ES consists of two steps, as shown in Figure 2 (b). In the first step, we predict multiple future trajectories based on the past trajectory and examples selected using past trajectory similarity, as outlined in Equation (1), yielding the prediction result $\hat{Y}_1 = \mathcal{F}(X, \mathcal{C})$. In the second step, these predicted future trajectories, $\hat{Y}_1$, are used to further refine the context selection, as shown in the following equation:

$$\mathcal{C}' = \text{Top-M}_{[\tilde{X}_1^i, \tilde{Y}_1^i] \in \mathcal{D}'^s} \left[ \min_{k \in \{1, \ldots, K\}} S\left( [X_1, \hat{Y}_1^k], [\tilde{X}_1^i, \tilde{Y}_1^i] \right) \right], \quad |\mathcal{C}'| = M. \tag{3}$$

The similarity is computed between the concatenated past trajectory and each of the multiple predicted future trajectories of the target agent, as well as the concatenated past and ground-truth future trajectory of the example. The minimum similarity value across all $K$ predictions is then used as the selection metric. By employing this two-step selection process, we can identify more relevant examples for in-context learning, taking into account both past and future trajectories.

### 3.4 Model Architecture

Our model consists of an embedding layer, a trajectory encoder, an in-context-aware trajectory predictor, and a multi-modal decoder, as presented in Figure 2 (a). Initially, the embedding layer $\mathcal{G}$ is applied to the agent's past trajectory to obtain $d$-dimensional features. We then concatenate $\mathcal{G}(X)$ with learnable query tokens $Q \in \mathbb{R}^{N \times T_{\text{pred}} \times d}$ and pass them through the trajectory encoders, Encoder to obtain the trajectory feature (*e.g.*, spatio-temporal features are extracted using a Social-Transmotion encoder [55]) as follows: $[H', Q'] = \text{Encoder}(\mathcal{G}(X), Q)$. Similarly, we use the embedding layer and trajectory encoder to obtain the trajectory features of $M$ in-context examples. Here, instead of employing learnable queries, we apply $\mathcal{G}$ to both the in-context past trajectories $X^i$ and future trajectories $Y^i$ for all $M$ examples: $[H^{i'}, Z^{i'}] = \text{Encoder}(\mathcal{G}(X^i), \mathcal{G}(Y^i))$, where $\quad i = 1, \ldots, M$.

Since the coordinates of the agents in the context examples are normalized to the position of each primary agent, it is crucial to integrate the relative position information of the target agent into the features of the in-context examples. We refer to this as the relative context position encoding (RCPE). RCPE is implemented using a simple one-layer MLP as follows: $\text{RCPE} = \text{MLP}(x_{rel}, y_{rel})$, where $(x_{rel}, y_{rel})$ represent the relative position of the primary agent of a context example with respect to the target agent. To incorporate information about which context example's primary agent is more similar to the target agent, we introduce similarity rank position encoding (SRPE), which encodes the similarity ranking of each context example's primary agent relative to the target agent. This is implemented using the original sinusoidal positional encoding from [61]. Both RCPE and SRPE are applied to the features of the context examples. These features are then fed into the in-context-aware trajectory predictor (Predictor), which consists of a three-layer Transformer encoder [61], allowing it to leverage the context examples for prediction as follows:

$$[\hat{H}_1, \hat{Q}_1, \hat{H}_1^1, \hat{Z}_1^1, \ldots, \hat{H}_1^M, \hat{Z}_1^M] = \text{Predictor}([H_1', Q_1',$$
$$H_1^{1'} + \text{RCPE}(1) + \text{SRPE}(1), Z_1^{1'} + \text{RCPE}(1) + \text{SRPE}(1), \ldots,$$
$$H_1^{M'} + \text{RCPE}(M) + \text{SRPE}(M), Z_1^{M'} + \text{RCPE}(M) + \text{SRPE}(M)]).$$
$$(4)$$

Finally, we implement the multimodal decoder, MultiModalDecoder, which consists of $K$ simple one-layer MLPs for multimodal prediction. The decoding process can be formulated as follows:

$$\hat{Y}_1 = \text{MultiModalDecoder}(\hat{Q}_1'), \quad \hat{Y}_1 \in \mathbb{R}^{K \times T_{\text{pred}} \times 2}. \tag{5}$$

### 3.5 Training and Inference

Our framework comprises two sequential training phases—vanilla trajectory prediction (VTP) training and in-context training—followed by an inference phase.

**Training.** The VTP training phase equips TrajICL with foundational trajectory prediction capabilities. Analogous to conventional trajectory prediction models, TrajICL learns to forecast the future trajectory of a target agent based on its historical path and the trajectories of surrounding agents. The subsequent in-context training phase enables TrajICL to perform effective in-context learning. During this phase, the model trains on examples where similar instances are selected using the proposed metric as detailed in Section 3.2. Both training phases utilize MSE loss, implementing a winner-take-all strategy that optimizes only the most accurate prediction: $\mathcal{L} = \min_k \|\hat{Y}_1^{(k)} - Y_1\|^2$. Note that we only employ a large-scale synthetic dataset to train the model.

**Inference.** During inference, we adopt the prediction-guided example selection from the scenario-specific samples introduced in Section 3.3, allowing the model to adapt to environmental changes and domain shifts without any additional parameter updates.

## 4 Experiments

### 4.1 Experimental Settings

**Datasets.** We train our model using the MOTSynth [14] dataset, a synthetic pedestrian detection and tracking dataset, consisting of over 700 90-second videos captured from various camera viewpoints in diverse outdoor environments. For our experiments, we use a subset of 424 scenes for training and 107 scenes for evaluation, all captured with a static camera. In addition to in-domain evaluation on MOTSynth, we assess our method on five widely adopted datasets for cross-domain evaluation: JRDB [45] (in both image coordinates, JRDB-Image, and world coordinates, JRDB-World), Wild-Track [8], SDD [53], and JTA [15]. It is important to note that the model is not trained on these datasets for cross-domain evaluation; only a few examples are used for inference. Following standard practice, we predict 12 future timesteps based on the previous 9 timesteps. For consistency with the human seconds protocol adopted by [60], we sort the $N$ identities chronologically based on their initial appearance in the scene. The earliest $80\%$ of identities are used to construct the example pool, while the remaining $20\%$ are reserved for evaluation.

**Evaluation Metrics.** We evaluate the performance of different trajectory prediction methods using two standard metrics: $\text{minADE}_K$ and $\text{minFDE}_K$. $\text{minADE}_K$ calculates the minimum average displacement error over time among the $K = 20$ predicted trajectories and the ground-truth future trajectory following the standard protocol [41, 22, 43]. $\text{minFDE}_K$ measures the minimum final displacement error, computing the distance between the closest predicted endpoint among the $K = 20$ predictions and the ground-truth endpoint.

**Baselines.** We combine TrajICL with the transformer-based model Social-Transmotion [55]. Although Social-Transmotion is designed to process multiple modalities, we use only trajectory data as input in all experiments to ensure fair comparisons. In addition, following recent works [9, 21, 18], we incorporate multiple forecasting heads to generate $K$ possible future predictions. To verify the effectiveness of our method, we compare it with versions of Social-Transmotion that have been fine-tuned using various methods, including full fine-tuning (Full FT), PEFT methods such as LoRA [24],

Table 1: Comparison with baseline methods on the MOTSynth, JRDB, WildTrack, SDD, and JTA datasets. minADE$_K$/minFDE$_K$ are reported. The unit for MOTSynth, WildTrack, and SDD is pixels, while the unit for JRDB-World and JTA is meters. **Bold** and underlined fonts represent the best and second-best results, respectively. The difference ($\Delta$) represents the percentage improvement achieved by TrajICL over the vanilla Social-Transmotion.

| Method | Training-free | In-Domain | Cross-Domain | | | | |
|---|---|---|---|---|---|---|---|
| | | MOTSynth | JRDB-Image | WildTrack | SDD | JRDB-World | JTA |
| Social-Transmotion [55] | ✓ | 17.6/23.0 | 2.88/3.32 | 24.7/36.3 | 10.2/18.9 | 0.15/0.26 | 1.18/1.97 |
| +Head Tuning | | 17.1/22.6 | 2.70/3.13 | 23.9/34.7 | 9.81/18.3 | 0.11/0.16 | 0.68/1.07 |
| +VPT Shallow [4] | | 17.8/23.5 | 2.80/3.33 | 24.3/37.4 | 8.73/14.8 | 0.11/0.16 | 0.61/0.92 |
| +VPT Deep [4] | ✗ | 17.7/23.9 | 2.81/3.24 | 24.4/37.9 | 8.84/15.6 | 0.10/0.15 | 0.60/0.90 |
| +LoRA ($r=16$) [24] | | 16.9/22.2 | 2.64/3.02 | 23.8/34.5 | 9.02/16.6 | 0.10/0.16 | 0.61/0.93 |
| +LoRA ($r=64$) [24] | | 16.8/22.2 | 2.65/2.98 | 23.6/35.6 | 9.11/16.8 | 0.10/0.16 | 0.60/0.93 |
| +Full FT | | 16.0/20.9 | **2.56**/2.87 | 22.9/34.5 | **7.96/13.6** | **0.09/0.14** | **0.52/0.76** |
| +TrajICL (Ours) | ✓ | **15.3/17.5** | 2.61/**2.68** | **21.1/28.3** | 8.40/14.8 | 0.13/0.21 | 0.59/0.85 |
| $\Delta$ | | -14.2%/-23.9% | -7.6%/-19.2% | -14.6%/-22.0% | -17.6%/-21.7% | -3.3%/-19.2% | -41.5%/-56.9% |

Table 2: Comparisons on JRDB-World and JTA under a few-shot evaluation setting. minADE$_K$/minFDE$_K$ are reported for different percentages of labeled real data available for the example pool for TrajICL and fine-tuning for the fine-tuning methods.

| Method | Training-free | MOTSynth | | JRDB-Image | | WildTrack | | SDD | | JTA | |
|---|---|---|---|---|---|---|---|---|---|---|---|
| | | 10% | 20% | 10% | 20% | 10% | 20% | 10% | 20% | 10% | 20% |
| Social-Transmotion [55] | ✓ | 17.6/23.0 | | 2.88/3.32 | | 24.7/36.3 | | 10.2/18.9 | | 1.18/1.97 | |
| +Head Tuning | | 17.5/23.0 | 17.4/22.9 | 3.00/3.50 | 2.84/3.25 | 24.5/35.9 | 24.5/35.2 | 10.2/19.1 | 10.2/19.1 | 0.78/1.32 | 0.76/1.23 |
| +VPT Shallow [4] | | 21.3/30.2 | 20.5/28.7 | 3.51/4.23 | 2.98/3.75 | 25.7/41.5 | 24.9/39.5 | 11.2/20.4 | 10.5/18.1 | 0.77/1.17 | 0.70/1.03 |
| +VPT Deep [4] | ✗ | 21.2/29.2 | 19.8/26.8 | 3.48/4.16 | 3.16/3.60 | 24.4/35.6 | 24.5/37.2 | 10.2/18.3 | 10.3/18.1 | 0.73/1.09 | 0.68/1.00 |
| +LoRA ($r=16$) [24] | | 17.4/23.0 | 17.4/22.9 | 2.94/3.54 | 2.75/3.24 | 24.0/35.0 | 23.9/34.0 | 10.1/19.0 | 10.1/18.8 | 0.75/1.22 | 0.70/1.14 |
| +LoRA ($r=64$) [24] | | 17.4/22.9 | 17.3/22.8 | 3.00/3.63 | 2.76/3.20 | 24.3/34.8 | 23.8/34.1 | 10.1/18.8 | 10.2/19.1 | 0.74/1.22 | 0.70/1.13 |
| +Full FT | | 17.0/22.2 | 16.8/21.8 | 3.30/4.56 | 2.78/3.16 | 24.5/39.4 | 23.4/34.3 | 10.6/18.6 | 9.83/17.2 | 0.68/1.09 | 0.64/0.99 |
| + TrajICL (Ours) | ✓ | **16.7/20.4** | **16.4/19.9** | **2.85/3.24** | **2.68/2.94** | **23.4/35.0** | **22.6/33.2** | **9.76/17.4** | **9.60/16.8** | **0.62/0.96** | **0.61/0.90** |

VPT [4], and head tuning, which adjusts the prediction heads. The data from the example pool is used for fine-tuning methods to ensure that all models have access to the same information.

**Implementation Details.** Our training process is divided into two stages: VTP training and in-context training. In the first stage, we train the model using the AdamW optimizer [40] with a base learning rate of $1 \times 10^{-3}$ for 100 epochs. We perform a 3-epoch warmup and decay the learning rate to 0 throughout training using the cosine annealing scheduler [39]. In the second stage, we train the model for 400 epochs, with a 12-epoch warmup and the cosine annealing scheduler, following the same setup as in the first stage. We set $M$ (number of in-context examples) to eight. The hyperparameters were determined through a standard coarse-to-fine grid search or step-by-step tuning. We set the batch size to 16 and train the model using one NVIDIA RTX A6000 GPU. The model configuration for Predictor consists of three layers and four attention heads, with a model dimension of $d = 128$. We employ Leaky ReLU functions as the activation function. Data augmentation techniques, including rotation, noise addition, and masking, as adapted from [18], are applied.

## 4.2 Comparison with Baseline Methods

Table 1 shows that TrajICL consistently outperforms Social-Transmotion by a significant margin across all datasets, highlighting its adaptability to a wide range of scenarios. On the MOTSynth, JRDB-Image, WildTrack, and SDD datasets, TrajICL exceeds the performance of the best fine-tuned methods, including full fine-tuning, in terms of minFDE$_K$. Furthermore, on the JRDB-World and JTA datasets—where pedestrian positions are provided in real-world coordinates—TrajICL remains competitive, demonstrating its generalizability across various sensor configurations. Despite being trained solely on synthetic data and without requiring any additional fine-tuning, these results showcase TrajICL's effectiveness in diverse scenarios, overcoming the third challenge of adaptability to new environments.

## 4.3 Comparison with Limited Pool Size

In Section 4.2, we validate the effectiveness of TrajICL using a sufficiently large example pool. However, in real-world scenarios, acquiring annotations manually incurs collection costs, and even when using detection and tracking algorithms, gathering a sufficient example pool takes time. Therefore,

Table 3: Ablation study of the prediction-guided STES for inference. $minFDE_K$ is reported. The subscript percentage denotes relative $minFDE_K$ reduction over random selection.

| Spatial | Temporal | Prediction-guided | MOTSynth | WildTrack | JTA | SDD |
|---|---|---|---|---|---|---|
| | | | 21.8 | 35.5 | 1.08 | 16.4 |
| | ✓ | ✓ | $18.0_{-17.4\%}$ | $\underline{28.9}_{-18.6\%}$ | $\mathbf{0.85}_{-21.3\%}$ | $\underline{15.1}_{-7.9\%}$ |
| ✓ | | ✓ | $\mathbf{16.7}_{-23.4\%}$ | $29.4_{-17.2\%}$ | $\underline{0.89}_{-17.6\%}$ | $15.9_{-3.0\%}$ |
| ✓ | ✓ | | $19.2_{-11.9\%}$ | $32.5_{-8.50\%}$ | $0.91_{-15.7\%}$ | $16.2_{-1.2\%}$ |
| ✓ | ✓ | ✓ | $\underline{17.5}_{-19.7\%}$ | $\mathbf{28.3}_{-20.3\%}$ | $\mathbf{0.85}_{-21.3\%}$ | $\mathbf{14.8}_{-9.8\%}$ |

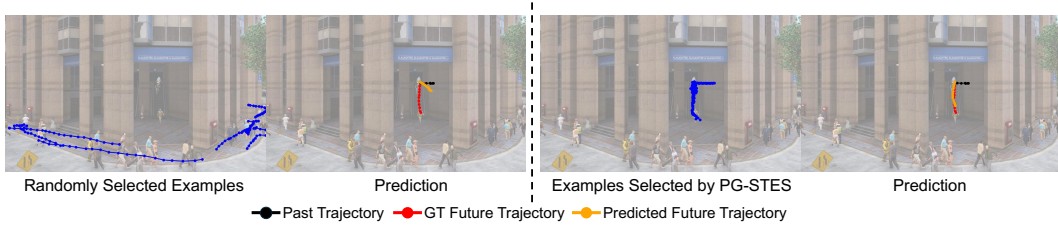

(a) MOTSynth  (b) JTA

Figure 3: Performance of random example selection and the proposed STES at varying numbers of in-context examples.

we evaluate the effectiveness of TrajICL with a limited pool size, which is a realistic and challenging scenario in real-world applications. We selected subsets of $5\%$ and $10\%$ from the example pool, using each subset as the example pool in TrajICL and as the fine-tuning data for the fine-tuning methods. The models were then evaluated on the same test set. As shown in Table 2, while PEFT methods demonstrate their effectiveness over full fine-tuning in the $100\%$ example setting (Section 4.2), our method consistently outperforms both Social-Transmotion and fine-tuning methods. In the most challenging scenario, with only $10\%$ of annotations, TrajICL achieves improvements of $8.8\%$, $7.0\%$, $4.9\%$, and $11.9\%$ on MOTSynth, JRDB-Image, SDD, and JTA, respectively, compared to the best-performing fine-tuning model in terms of $minFDE_K$. The results of ETH-UCY [49, 30] and NBA SportVU [74] are provided in the supplementary.

## 4.4 Ablation Studies

**Effectiveness of STES**. We first evaluate the impact of incorporating our STES into in-context learning. As shown in Figure 3, STES consistently results in improvements as the number of examples increases across different datasets. In contrast, randomly selected examples yields minimal performance gains, even as the number of examples increases in the MOTSynth dataset. In JTA, no improvement is observed despite the increase in examples. These results highlight the effectiveness of STES, which efficiently equips the model with ICL capabilities, overcoming the first limitation.

**Effectiveness of the PG-STES**. We conduct ablation experiments to evaluate the impact of our proposed PG-STES on inference. As shown in Table 3, PG-STES consistently outperforms random selection in terms of $minFDE_K$ across all datasets. By incorporating both the past trajectory and the predicted future trajectory for example selection, PG-STES improves over STES, which only uses past trajectory for selection, by $8.1\%$, $7.4\%$, $6.6\%$, and $8.6\%$ on the MOTSynth, WildTrack, JTA, and SDD datasets in terms of $minFDE_K$, respectively. These results demonstrate the effectiveness of combining predicted future trajectories with past trajectories to enhance performance, addressing the second limitation. We also investigate the impact of spatial and temporal components in the STES

Randomly Selected Examples · Prediction · Examples Selected by PG-STES · Prediction

—— Past Trajectory  —— GT Future Trajectory  —— Predicted Future Trajectory

Figure 4: Qualitative comparison between random example selection and our proposed PG-STES.

| Table 4: Ablation study of first-stage VTP training. $\text{minFDE}_K$ is reporeted. | | | |
|---|---|---|---|
| VT | MOTSynth | WildTrack | JTA |
|  | 15.4/17.6 | 21.9/30.4 | 0.60/0.89 |
| ✓ | **15.3/17.5** | **21.1/28.3** | **0.59/0.85** |

Table 5: Ablation study of RCPE and SRPE.

| RCPE | SRPE | MOTSynth | WildTrack | JRDB-World | JTA |
|---|---|---|---|---|---|
|  |  | 15.5/18.1 | 21.9/31.0 | 0.14/0.23 | 0.61/0.91 |
| ✓ |  | 15.4/17.6 | 21.8/29.9 | 0.14/0.24 | 0.59/0.88 |
|  | ✓ | **15.3**/17.6 | 21.8/29.6 | 0.14/0.23 | 0.59/0.89 |
| ✓ | ✓ | **15.3/17.5** | **21.1 /28.3** | **0.13/0.21** | **0.59/0.85** |

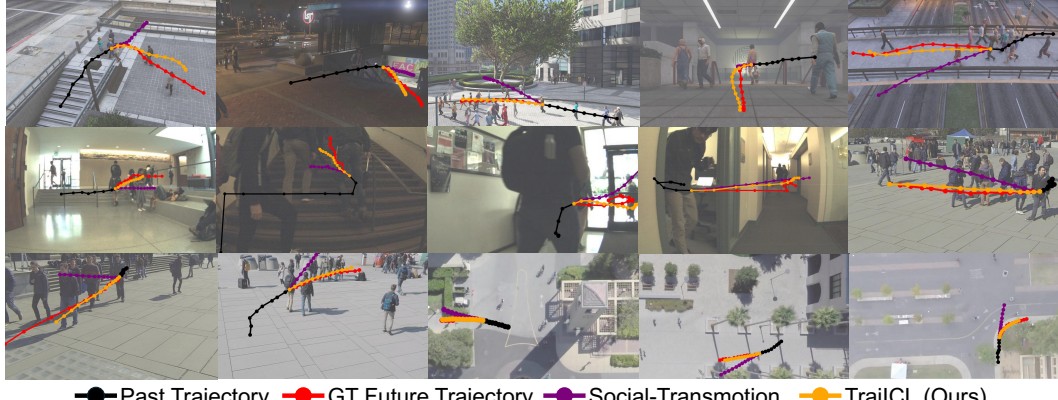

▬●▬Past Trajectory  ▬●▬GT Future Trajectory  ▬●▬Social-Transmotion  ▬●▬TrajICL (Ours)

Figure 5: Qualitative results on MotSynth, JRDB, WildTrack, and SDD. These examples demonstrate scenarios where our TrajICL outperforms the Social-Transmotion baseline. TrajICL effectively learns the plausible motion patterns from examples.

example selection process. The results indicate that temporal similarity yields greater improvements than spatial similarity on the JTA and SDD datasets. However, on the MOTSynth dataset, spatial similarity provides more substantial gains. This suggests that the optimal selection dimension for demonstration examples is dataset-dependent. When both spatial and temporal similarities are utilized together, our method achieves the best performance across most datasets, particularly when prediction-guided retrieval is applied.

**Effectiveness of VTP Training.** Our experiments demonstrate that VTP training consistently delivers performance improvements across multiple datasets, as shown in Table 4.

**Effectiveness of RCPE and SRPE.** We investigate the impact of RCPE, which encodes the relative position with respect to the target agent, and SRPE, which encodes the similarity rank of the context examples' primary agents in relation to the target agent. As shown in Table 5, our ablation study demonstrates that both RCPE and SRPE contribute to improved performance across various datasets. On the in-domain MOTSynth dataset, RCPE alone achieves a slight improvement in performance. However, on the cross-domain datasets, WildTrack and JRDB-World, the combination of both RCPE and SRPE results in more significant improvements. Further ablation studies and analysis are available in the supplementary material.

**Effect of Architecture.** Even a weak base model with a shallow encoder and small dimensions exhibits ICL capabilities, as shown in Tables 6 and 7. While a deeper encoder and larger dimensions offer negligible benefits for the zero-shot case, they significantly boost performance in ICL.

| Table 6: Effect of model dimension on MotSynth. $\text{minFDE}_K$ is reporeted. | | |
|---|---|---|
| Dim. | zero-shot | 8-shots |
| 32 | 19.2 | 17.7 |
| 64 | **17.9** | 16.5 |
| 128 | 18.1 | **15.3** |

| Table 7: Effect of predictor depth on MotSynth. $\text{minFDE}_K$ is reporeted. | | |
|---|---|---|
| Depth | zero-shot | 8-shots |
| 1 | 19.1 | 16.6 |
| 2 | 18.8 | 16.9 |
| 3 | **18.1** | **15.3** |

Table 8: Performance of TrajICL with ForecastMAE backbone. $\text{minFDE}_K$ is reporeted.

|  | MotSynth | JTA |
|---|---|---|
| ForecastMAE | 18.4 | 0.76 |
| +TrajICL (Ours) | **16.1** | **0.68** |
| Δ | -12.5% | -10.5% |

**Result of Different Backbone.** We selected Social-Transmotion as our backbone due to its state-of-the-art performance and generalized architecture. Our TrajICL framework is then integrated with this backbone to enable ICL capabilities. To confirm the framework's generalizability, we replaced the backbone with ForecastMAE [9] (disabling its lane encoder for pedestrian prediction). The results, summarized in Table 8, verify that our framework is generalizable and can be successfully applied to different backbones.

## 4.5 Qualitative Results

We compare randomly selected examples with those chosen by our PG-STES in Figure 4, along with their prediction results. PG-STES effectively selects spatially and temporally similar examples, enabling our model to generate more plausible predictions, such as a pedestrian riding down an escalator, with a better understanding of 3D structures compared to random selection. Figure 5 highlights the qualitative results of TrajICL and Social-Transmotion across various datasets, showcasing our method's adaptability in predicting future trajectories across domains. Unlike the Social-Transmotion baseline, which often predicts pedestrians floating in the air, our model aligns closely with the ground truth, even on non-planar surfaces like stairs. Furthermore, our approach incorporates finer-grained map awareness, avoiding obstacles like trees and respecting constraints (*e.g.*, not crossing fences), while capturing behavioral trends such as walking on sidewalks instead of roads.

## 5 Conclusion

In this paper, we introduce TrajICL, a novel in-context learning (ICL) framework for pedestrian trajectory prediction that enables adaptation without the need for fine-tuning on the domain-specific data. We address the challenges of incorporating ICL into trajectory prediction by employing spatio-temporal similarity-based example selection, prediction-guided example selection, and leveraging a large-scale synthetic trajectory dataset. In our experiments, we thoroughly validate that our approach effectively adapts to environmental variations and domain shifts. Despite these promising results, there remains work to be done. While increasing the number of in-context examples improves accuracy, it also raises computational costs during inference. We plan to explore this further in our future work.

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

# A  Appendix

## A.1  Implementation Detail

MOTSynth and WildTrack datasets are processed at 2.0 FPS, the 9-step observed past corresponds to 4.5 seconds, and the 12-step predicted future to 6.0 seconds. JRDB, JTA, and SDD datasets are processed at 2.5 FPS; these durations are 3.6 seconds for the past and 4.8 seconds for the future, respectively.

## A.2  Experiments on Additional Datasets

We also provide the results on ETH-UCY [49, 30] and NBA SportVU [74]. For NBA SportVU, we extracted two sub-datasets to use as benchmarks, named Rebounding and Scoring subsets, following the previous methods [71, 12]. Table 9 presents a performance comparison on the ETH-UCY. Across all subsets of ETH-UCY, TrajICL consistently surpasses Social-Transmotion by a notable margin and achieves competitive performance compared to fine-tuning methods, demonstrating its strong adaptability and robustness in diverse scenarios. Furthermore, we present a performance comparison on the challenging NBA SportVU dataset [74] in Table 10. When the full $100\%$ data pool is available, TrajICL outperforms Social-Transmotion in both subsets. However, on the rebounding subset, Social-Transmotion achieves better performance over TrajICL in terms of $minFDE_K$ when only $10\%$ of the pool is available. The reason is that the original scene segments in the NBA dataset are short, so the default pool size is already small. Maintaining strong performance even with a small data pool is crucial, and we plan to explore this further in our future work.

Table 9: Comparison with baseline methods on the ETH-UCY [49, 30]. $minADE_K$/$minFDE_K$ are reported. The unit is meters. **Bold** and underlined fonts represent the best and second-best results, respectively. The difference ($\Delta$) represents the percentage improvement achieved by TrajICL over the vanilla Social-Transmotion.

| Method | Training-free | ETH | HOTEL | UNIV | ZARA1 | ZARA2 | AVG |
|---|---|---|---|---|---|---|---|
| Social-Transmotion [55] | ✓ | 0.42/0.79 | 0.11/0.19 | 0.33/0.59 | 0.30/0.58 | 0.26/0.46 | 0.28/0.52 |
| +Head Tuning | | 0.46/0.85 | 0.17/0.31 | 0.30/0.53 | 0.25/0.47 | 0.23/0.42 | 0.28/0.52 |
| +VPT Shallow [4] | | 0.43/0.82 | 0.11/0.17 | 0.27/0.46 | 0.21/0.40 | 0.19/0.31 | 0.24/0.43 |
| +VPT Deep [4] | ✗ | 0.46/0.78 | 0.09/**0.13** | 0.25/0.44 | **0.19**/0.35 | **0.17**/0.29 | 0.23/0.40 |
| +LoRA ($r = 16$) [24] | | 0.38/0.78 | 0.09/**0.13** | 0.24/0.40 | **0.19**/0.36 | **0.17**/0.30 | **0.21**/0.39 |
| +LoRA ($r = 64$) [24] | | 0.44/0.78 | 0.09/**0.13** | 0.25/0.43 | 0.20/0.37 | **0.17**/0.30 | 0.23/0.40 |
| +Full FT | | 0.36/**0.64** | 0.09/**0.13** | 0.22/**0.39** | **0.19**/0.37 | 0.18/0.32 | **0.21**/**0.37** |
| +TrajICL (Ours) | ✓ | **0.34**/**0.64** | 0.10/**0.13** | 0.28/0.48 | 0.21/0.40 | 0.24/0.40 | 0.23/0.41 |
| $\Delta$ | | -19.0%/-9.1% | -7.6%/-31.6% | -15.1%/-18.6% | -36.7%/-31.0% | -7.7%/-13.0% | -17.9%/-21.2% |

Table 10: Comparisons on NBA SportVU. $minADE_K$ is reported for different percentages of labeled real data available for the example pool for TrajICL.

| Method | Training-free | Rebounding Subset | | Scoring Subset | |
|---|---|---|---|---|---|
| | | 10% | 100% | 10% | 100% |
| Social-Transmotion [55] | ✓ | 1.03/1.57 | | 1.02/1.91 | |
| + TrajICL (Ours) | ✓ | 0.93/1.61 | 0.90/1.35 | 0.98/1.78 | 0.94/1.73 |

## A.3  Comparison with Adapting Trajectory Prediction Methods

Most existing adaptive trajectory prediction methods exhibit tight coupling between the adaptation module and the prediction model architecture [60, 12, 34]. Moreover, these methods often employ varied training settings, such as different datasets and target domains, which complicates a fair, direct comparison. The evaluations in these prior works are also typically limited to baselines or commonly used prediction models. To establish a direct comparison, our main experiments include the conceptual methodology from a recent adaptation paper [60] (see Table 1 in the main paper, "+VPT Shallow"). This was achieved by integrating their visual prompt tuning approach into our Social-Transmotion base model. For additional context, the performance of other recent adaptive methods [12, 34] on the same evaluation datasets is Table 11. The results indicate that our

Table 11: Comparison with adaptive trajectory prediction methods. $\text{minADE}_K$ is reported.

| Model | SDD | NBA (Rebounding) |
|---|---|---|
| Latent Corridors [60] | 8.73 | - |
| RAN [12] | 11.0 | 1.28 |
| MetaTra [34] | 10.1 | - |
| Ours | **8.40** | **0.92** |

Table 12: Different prompt selection methods. $\text{minFDE}_K$ is reported.

| Selection Method | SDD | MotSynth |
|---|---|---|
| Random [20, 79] | 16.6 | 21.8 |
| Feature Sim. [2, 27] | 16.0 | 19.2 |
| Motion Sim. [65] | 19.2 | 32.5 |
| PG-STES (Ours) | **14.8** | **17.5** |

approach outperforms these methods. However, we note that a direct comparison is challenging due to significant differences in their underlying model architectures and training datasets.

## A.4 Detailed Ablation Study

**Effectiveness of the PG-STES.** We compare our PG-STES with in-context sample selection strategies from other computer vision fields in Table 12. The results confirm that our selection strategy achieves the best performance on the trajectory prediction task.

**Effectiveness of the STES.** Figure 6 demonstrates the impact of integrating our STES into ICL on the WildTrack SDD, JRDB-World, and JRDB-Image datasets. The inclusion of in-context examples consistently improves accuracy across all datasets, underscoring the effectiveness of STES in efficiently enhancing the model's ICL capability. Furthermore, as shown in Figure 7, we validate the effectiveness of STES with a larger number of examples. While STES continues to yield improvements as the number of examples increases across different datasets, the accuracy gains tend to saturate beyond a certain point.

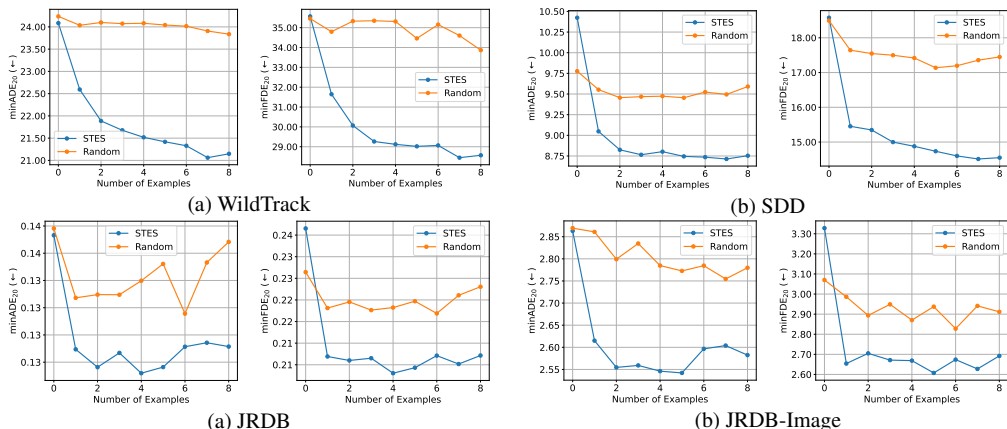

Figure 6: Performance of random example selection and the proposed STES at varying numbers of in-context examples.

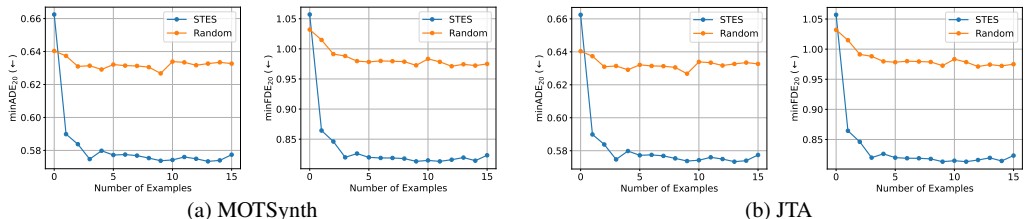

Figure 7: Performance of random example selection and the proposed STES at varying numbers of in-context examples with a larger number of examples.

**Effect of Pool Size.** We next investigate the impact of varying the size of the in-context pool. As shown in Figure 8, experiments on MotSynth reveal that increasing the number of examples

in the in-context pool leads to improved performance. TrajICL consistently outperforms all fine-tuning methods on MotSynth. On the other hand, for JTA, when the in-context pool is small, TrajICL surpasses the fine-tuning methods by effectively preventing overfitting, as it does not require additional parameter updates. However, as the pool size increases, full fine-tuning begins to achieve better results on JTA.

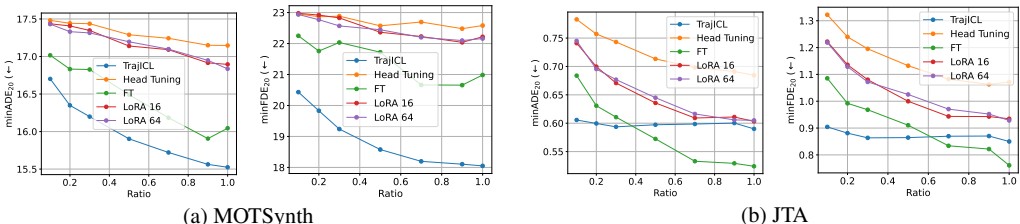

|  |  |
|:--:|:--:|
| (a) MOTSynth | (b) JTA |

Figure 8: Performance change brought by different sizes of the in-context pool.

## A.5 Analysis

**Adaptation and Inference Cost.** Our primary motivation is to eliminate the fine-tuning and back-propagation costs on edge devices. The backpropagation process requires significantly more GPU memory than inference, and our approach also mitigates the burdens of model management and environment-specific data collection. Therefore, in Table 13, we compare the fine-tuning cost of the base model with our method's inference cost. Our required GPU memory is significantly less than that needed for fine-tuning the base model, which aligns with our paper's primary motivation. While optimizing inference cost is not the main focus of our work, we report these figures for completeness in Table 14. We follow the experimental setup of existing ICL approaches [80], where features for in-context samples are pre-computed. Our overall inference cost is higher due to the additional Predictor module for aggregating in-context sample features and the two-stage inference via PG-ES. As mentioned in the conclusion, reducing the inference cost is a current limitation for edge-device deployment. We believe future work could explore existing techniques, such as Token Merging [6], to improve inference speed without requiring architectural modifications. The inference cost was computed on a machine with an Intel Xeon W-3235 CPU, 128GB of RAM, and an NVIDIA Titan RTX GPU, with GPU memory measured using a batch size of one.

Table 13: Comparison of adaptation cost.

| Model | FLOPs (GFLOPS) | GPU Memory (MB) |
|---|---|---|
| Social-Transmotion Fine-tuning | 10.43 | 462 |
| +TrajICL | 4.54 | 288 |

Table 14: Comparison of inference time.

| Model | FLOPs (GFLOPS) | Params (million) | Total Inference time (ms) |
|---|---|---|---|
| Social-Transmotion | 3.48 | 3.10 | 6.11 |
| +TrajICL | 4.54 | 4.15 | 7.69 + 0.04 (retrieval) |

Table 15: Robustness evaluation against real-world errors on JRDB-Image using inputs from the upstream detector and tracker perception modules.

| Model | Input | minADE$_{20}$ | minFDE$_{20}$ |
|---|---|---|---|
| Social-Transmotion | GT | 2.88 | 3.32 |
| Social-Transmotion | Off-the-shelf | 3.25$_{+12.8\%}$ | 4.33$_{+30.8\%}$ |
| +TrajICL (Ours) | GT | 2.61 | 2.68 |
| +TrajICL (Ours) | Off-the-shelf | **2.62**$_{+0.4\%}$ | **2.79**$_{+4.1\%}$ |

Table 16: Performance comparison of selection strategies with short observation trajectories on MOTSynth.

| Model | 9 Timestep | 6 Timestep | 3 Timestep |
|---|---|---|---|
| Social-Transmotion | 17.6/23.0 | 20.1/27.4 | 24.3/34.1 |
| +TrajICL w/ Random Selection | 16.6/21.8 | 19.0/23.6 | 19.3/25.7 |
| +TrajICL w/ STES Selection | 15.6/19.2 | 16.9/20.7 | 17.9/22.9 |
| +TrajICL w/ PG-ES Selection | **15.3/17.5** | **16.0/18.0** | **16.4/19.3** |
| Δ | -14.2%/-23.9% | -20.4%/-34.3% | -32.5%/-43.9% |

**Does ICL Offer Robustness to Real-World Perception Errors?** While improving robustness to trajectory noise is a separate line of research [18, 72], relying on clean, ground-truth trajectories is not feasible in real-world settings. Therefore, we conducted an additional experiment using trajectories generated by off-the-shelf models (Faster R-CNN [52] for person detection and Deep-SORT [66] for tracking, pretrained on the MOT17 dataset [46]) on the JRDB Image dataset. The results are summarized in Table 15. The "Input" column specifies whether trajectories are ground-truth (GT) or predicted by off-the-shelf models. The percentage value $(+X\%)$ indicates the performance

degradation when using predicted trajectories instead of GT ones. This setup simulates a fully automated pipeline that eliminates the need for manual annotation. The results demonstrate that while the baseline model's performance severely degrades on noisy inputs (e.g., $30.4\%$ increase in $\text{minFDE}_K$), our model maintains its robustness, with only a $4.1\%$ degradation in $\text{minFDE}_K$.

**Does ICL Performance Hold with Short Observation Trajectories?** We conducted an additional experiment using fewer past timesteps to evaluate performance on short observation trajectories. As summarized in Table 16, while absolute performance degrades with shorter inputs, our proposed selection strategies (especially PG-ES) still significantly outperform the random baseline. For instance, with only 3 timesteps, PG-ES reduces the prediction error from 25.7 (Random) to 19.3, demonstrating the effectiveness of our approach even with limited input data.

**How Does the Iterative Application of PG-ES Affect Trajectory Prediction Performance?** Better performance can be achieved by applying the method multiple times (at least twice), as shown in Table 17. However, as also mentioned, this increases the computational cost, since each iteration involves both sample retrieval and the feed-forward computation of the Predictor.

Table 17: Performance improvement via iterative application of PG-ES.

| Iteration Times | 1 | 2 | 3 | 4 |
| --- | --- | --- | --- | --- |
| $\text{minFDE}_K$ | 17.5 | 16.6 | 16.7 | 16.7 |

**What is the Relationship between Initial Prediction Error and the Effectiveness of the PG-ES Method?** To better understand the conditions under which PG-ES provides the most benefit, we analyze its performance gain relative to the initial prediction error. We measure the improvement of PG-ES over the STES as the improvement in $\text{minADE}_K$ in Figure 9. The results demonstrate PG-ES is most impactful on trajectories with high initial errors because such cases typically represent ambiguous, multimodal scenarios where a single past leads to multiple plausible futures. By generating a diverse set of hypotheses, the model can then correct its initial, erroneous prediction by selecting the most viable outcome. This correction process naturally yields the most significant improvements for the most challenging predictions.

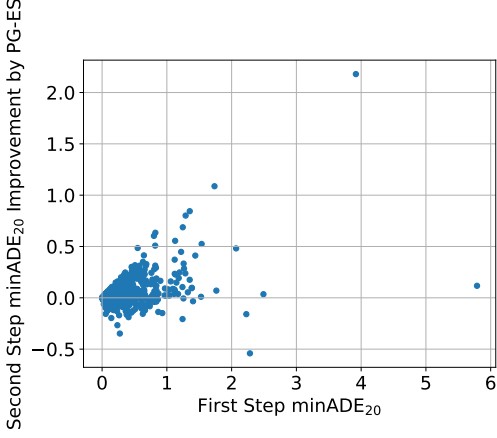
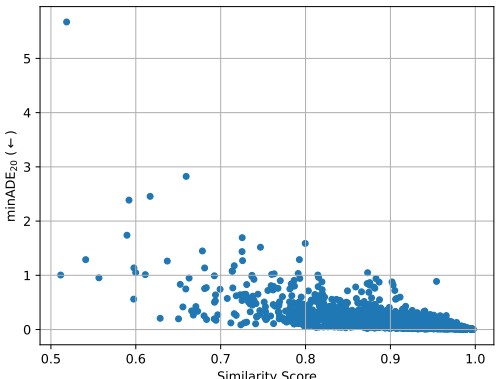

Figure 9: Impact of initial prediction error on the efficacy of PG-ES.

Figure 10: Effect of in-context similarity on performance

**Effect of Pool Quality on Prediction Accuracy.** To analyze the impact of in-context pool quality on prediction accuracy, we measured the similarity between a target trajectory and the examples in its pool using the STES metric. A low maximum similarity indicates that the pool lacks relevant examples for the target, as shown in Figure 10. The results show that performance degrades when the target trajectory has low similarity to the pool examples, which often corresponds to out-of-distribution (OoD) motion patterns. When we categorized target trajectories into three levels based on this similarity (low, medium, and high) in terms of $\text{minADE}_K$, Social-Transmotion achieved respective scores of $37.0/12.1/5.45$ (low/medium/high). In comparison, our TrajICL framework performed consistently better, achieving scores of $30.3/10.0/4.69$. Notably, even for the most challenging

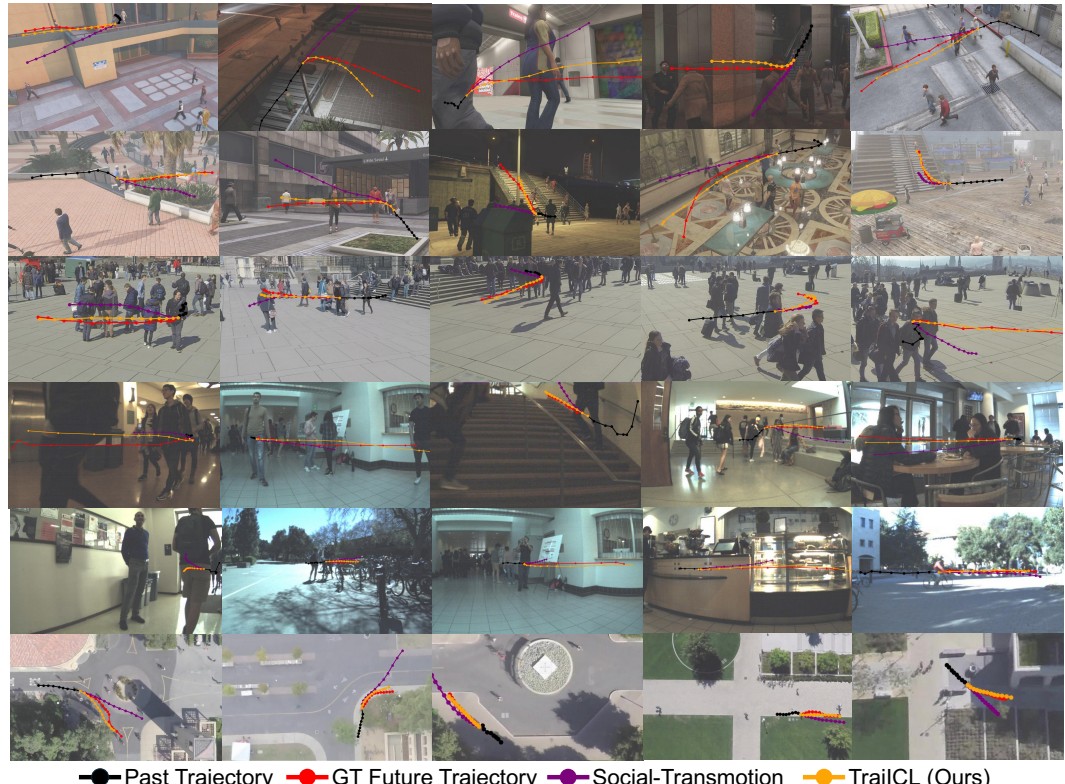

Figure 11: Qualitative results on MotSynth, JRDB, WildTrack, and SDD.

OoD inputs in the low-similarity group, our TrajICL framework shows a significant performance improvement over the baseline.

**What is the Key Component for Effective Generalization to Real-World Data?** Although our synthetic training data has limited motion diversity compared to real-world datasets, our model generalizes effectively to challenging benchmarks like WildTrack, JRDB, and SDD. This strong generalization is primarily due to our PG-STES module. During inference, rather than relying solely on the synthetically trained model, PG-STES dynamically selects examples relevant to the target trajectory from a pool of test data. This process provides the model with in-domain knowledge of the current scene, effectively bridging the synthetic-to-real domain gap. This approach is validated in Table 3 of our main paper, where using randomly selected in-context samples fails to improve performance on the JTA dataset. To further quantify the effectiveness of our selection strategy, we measured the Fréchet Inception Distance (FID) between the feature distributions of selected in-context examples and the synthetic training data. Our PG-STES method achieves an FID score of $612$, a stark contrast to the $2,454,212$ score from random selection. This substantially lower FID score demonstrates that our strategy is highly effective. By providing the model with the most pertinent real-world examples at inference time, PG-STES enables robust generalization and overcomes the limitations of synthetic training.

## A.6   More Qualitative Results

We present further qualitative comparisons of TrajICL and Social-Transmotion on the MOTSynth, JRDB-Image, WildTrack, and SDD datasets in Figure 11. Compared to the Social-Transmotion baseline, our model demonstrates closer alignment with the ground truth by incorporating finer-grained map awareness and effectively avoiding obstacles.

