# OpenReview forum: "Towards Predicting Any Human Trajectory In Context"
_NeurIPS.cc/2025/Conference — NeurIPS 2025 poster_

### Official Review · Reviewer_8wc2 · 2025-06-29

**Clarity:** 3
**Significance:** 4
**Originality:** 3
**Rating:** 5
**Confidence:** 4

**Summary:**

The paper proposes a novel in-context learning (ICL) model for pedestrian trajectory prediction, showing improved results compared to the base model in novel environments without additonal fine-tuning.

**Questions:**

1. Can the authors provide an analysis of the inference speed of the proposed method, especially how it compares to the speed of the base model without the ICL modules?
2. Can the authors show that the proposed ICL module is general to different base models, such as a more lightweight one?
3. Can the authors explain how the proposed example selection strategies could be incorporated into real-world settings where no ground truth annotations are available?

**Ethical Concerns:**

["NO or VERY MINOR ethics concerns only"]

**Final Justification:**

The authors addressed my concerns in the rebuttal. The additional results on noises from pseudo-labels further highlight the significance of the proposed ICL for trajectory prediction in real-world settings. I suggest accepting this work.

**Limitations:**

Trajectory prediction, particularly in an in-context learning setting, may raise some ethical concerns.

**Quality:**

3

**Strengths And Weaknesses:**

Strengths:
1. The motivation is clear, and the paper is easy to follow.
2. The idea of applying ICL to trajectory prediction is novel, and the results are strong and promising.
3. Ablation studies are extensive, clearly showing the effectiveness of each component.

Weaknesses:
1. The synthetic dataset MOTSynth used to train the ICL model is an established dataset and should not be counted as a key contribution.
2. The motivation for using ICL is to reduce the costly fine-tuning for easy deployment on edge devices; however, the base model and the addition of ICL modules all seem quite computationally expensive based on the transformer architecture, limiting their potential in real-world applications.
3. The paper only experiments on datasets with ground-truth annotations. However, a real-world ICL system involves first collecting and annotating a few real-world examples instead of annotating a large number of samples and then selecting from them, as the latter sounds too laborious and conflicts with the motivation of ICL. I'm concerned that the proposed example selection approaches may not be informative and have enough significance in the real world.

---

> ### Author Rebuttal · Authors · 2025-07-31
>
> Dear Reviewer 8wc2,
>
> We sincerely appreciate your insightful feedback. We have addressed the concerns raised in the review as detailed below.
>
> **Weakness 1: Using synthetic dataset as a contribution**
>
> We agree that simply using an existing dataset like MOTSynth is not a key contribution of the paper.
>
> However, we want to emphasize a crucial finding: unlike previous approaches [1, 2, 3] that rely heavily on curating real-world trajectory datasets for training, our paper shows that a trajectory prediction model can be trained exclusively on synthetic data and performs well by using real-world data as in-context samples. This is a significant finding for the community, as it demonstrates a way to reduce the costly burden of real-world data curation.
>
> To avoid confusion, we will remove this from our list of contributions.
>
> [1] Social-Transmotion: Promptable Human Trajectory Prediction. In ICLR 2024.
>
> [2] Trajectron++: Dynamically Feasible Trajectory Forecasting with Heterogeneous Data. In ECCV 2020.
>
> [3] Physical plausibility-aware trajectory prediction via locomotion embodiment. In CVPR 2025.
>
> **Weakness 2, Question 1: Inference speed analysis**
>
> First, we would like to emphasize that our main motivation is to eliminate the fine-tuning and backpropagation costs on edge devices since the backpropagation process requires much higher GPU memory compared to the inference, as well as the burden of model management and environment-specific data collection (L.4, L.31-34, L.41...). This is what we refer to as "efficiency." We did not aim to accelerate inference time in the original paper.
>
> We will revise the term "efficiency" in the final manuscript to avoid confusion.
> However, as you pointed out, it is important to analyze the inference cost and compare it with the base model.
>
> | Model | FLOPs (GFLOPS) | Params (million) | Total Inference time (ms) |
> | ----  | ---- | ----  | ---- |
> | Social-Transmotion| 3.48  | 3.10 |6.11 |
> | TrajICL | 4.54  | 4.15  |7.69 + 0.04 (retrieval) |
>
> (Inference time measure with PC equipped with NVIDIA TITAN RTX GPU, 188 GB RAM, and Intel(R) Xeon(R) Silver 4216 CPU.)
>
> We follow the experimental setup of existing in-context learning approaches [1], where features for in-context samples are pre-computed. Our overall inference cost is higher because of the additional Predictor module for aggregating in-context sample features and the two-stage inference via PG-ES.
>
> However, since our primary motivation is to eliminate the training step after deployment, we also compared the fine-tuning cost of the base model with our inference cost:
>
> | Model | FLOPs (GFLOPS) |  GPU Memory (MB) |
> | ----  | ----  | ---- |
> | Social-Transmotion Fine-tuning | 10.43 | 462 |
> | TrajICL | 4.54  |288 |
>
> (GPU memory is computed with batch size = 1)
>
> Our required GPU memory is significantly less than that needed for fine-tuning the base model, which aligns with our paper's primary motivation. As mentioned in the conclusion, reducing the inference cost is a current limitation for edge-device deployment. We believe future work could explore applying existing techniques, such as Token Merging [2], to improve inference speed without requiring architectural modifications.
>
> [1] In-Context Learning Makes an Ego-evolving Scene Text Recognizer, In CVPR 2024.
>
> [2] Token Merging: Your ViT But Faster, In ICLR 2023.
>
>
> **Question 2: Performance of different base models**
>
> As you suggested, we trained a lighter model with a reduced feature dimension size (from 128 to 64) without VTP on MOTSynth. Due to time constraints during the rebuttal, we experimented with a 4-shot model.
>
> | | ADE| FDE |
> | ----  | ---- | ---- |
> | zero-shot |  16.41 | 19.36 |
> | 4-shot |  **15.30** | **17.50** |
>
> This table shows that in-context learning capabilities are still acquired with a lighter base model.
>
> Additionally, to further confirm the generalization capability, we changed the backbone to ForecastMAE [1] (disabling the lane encoder for pedestrian prediction) and conducted experiments. The results are summarized below:
>
> | Model | MOTSynth | WildTrack | JTA|
> | ----  | ---- | ----  | ---- |
> | ForecastMAE | 18.41/24.51  | 25.0/29.5 |0.76/1.23 |
> | ForecastMAE + TrajICL (4 shots) | **15.98**/**17.91**  |  **21.9/28.3**  | **0.64**/ **0.91** |
>
> These results verify that our framework is generalized and can be applied to different backbones.
>
> [1] Forecast-MAE: Self-supervised Pre-training for Motion Forecasting with Masked Autoencoders, In ICCV 2023.
>
> **Weakness 3, Question 3: Robustness to Real-world noises**
>
> Thank you for this insightful feedback. While improving robustness to trajectory noise is a separate line of research [1, 2, 3], we agree that using clean, ground-truth trajectories is not feasible in a real-world setting. Therefore, we conducted an additional experiment using trajectories generated by off-the-shelf models (Faster R-CNN [4] for person detection and Deep-SORT [5] for tracking pretrained on the MOT17 dataset) on the JRDB Image dataset. The results are summarized below.
>
> | Model | Input | ADE &darr;	 | FDE &darr; |
> | ---- | ---- | ---- | ---- |
> | Social-Trans| GT | 2.88 |  3.32 |
> | Social-Trans | Off-the-shelf | 3.25 (+12.8%) | 4.33 (+30.4%) |
> | TrajICL (Ours) | GT | 2.61 | 2.68 |
> | TrajICL (Ours) | Off-the-shelf | 2.62 (+0.4%) | 2.79 (+4.1%) |
>
> The "Input" column indicates whether the target and in-context samples are ground-truth (GT) or are predicted by off-the-shelf models. The (+X%) value shows the performance change from using GT trajectories to using predicted trajectories. This setup represents a fully automated process that doesn't require manual ground-truth annotation.
>
> The table shows that while the zero-shot model's performance severely degrades with noisy inputs (e.g., +30.4% in FDE), our model robustly predicts future trajectories from noisy inputs, with only a 4.1% degradation in FDE.
>
> [1] RealTraj: Towards Real-World Pedestrian Trajectory Forecasting, In arXiv 2025.
>
> [2] Uncovering the missing pattern: Unified framework towards trajectory imputation and prediction, In CVPR 2023.
>
> [3] BCDiff: Bidirectional Consistent Diffusion for Instantaneous Trajectory Prediction, In NeurIPS 2023.
>
> [4] Faster R-CNN: Towards Real-Time Object Detection with Region Proposal Networks. In NeurIPS 2025.
>
> [5] Simple Online and Realtime Tracking with a Deep Association Metric, In ICIP 2017.
>
> **Limitation: Ethical Concerns**
>
> We would like to kindly ask the reviewer to elaborate on the ethical considerations of the trajectory prediction with in-context learning so that we will incorporate that in the final manuscript.

---

> ### Comment · Reviewer_8wc2 · 2025-08-01
>
> Thank the authors for their informative and detailed rebuttal, which addressed my concerns. I will raise my rating to accept. I appreciate the results presented in the response to question 3, which reveal another advantage of ICL in being robust to real-world label noises, in addition to lower training costs. I suggest including a more comprehensive version of this experiment in the main paper.
>
> For the minor ethical concerns, though I think it might not be significant for additional ethical reviews, I do believe the method could potentially be exploited in surveillance applications like predicting the trajectory of certain people using few-shot examples.

---

> ### Author Response · Authors · 2025-08-02
> **Authors' Response**
>
> Dear reviewer 8wc2,
>
> We sincerely appreciate your quick action in reading our rebuttal and raising your score.
> As you suggested, we will revise the final manuscript accordingly:
>
> - Include a more comprehensive version of the robustness-to-noise experiment in the main paper to highlight the additional advantages of ICL.
> - Add a section in the ethical considerations to discuss the potential misuse of trajectory prediction models in surveillance scenarios.
> - Remove the contribution of using the synthetic dataset.
> - Include other experiments (e.g., different backbones, inference speed) in either main paper or supplementary..
>
> Once again, thank you for your thoughtful feedback and for taking the time to help improve our work.
>
> Best regards,

---

### Official Review · Reviewer_od3z · 2025-07-01

**Clarity:** 3
**Significance:** 3
**Originality:** 4
**Rating:** 4
**Confidence:** 4

**Summary:**

The paper presents TrajICL, an in-context learning framework that does not require fine-tuning for human trajectory prediction. To identify relevant in-context examples, the approach incorporates a two-stage selection mechanism based on spatio-temporal similarity and prediction-guided refinement. The model is trained on a large-scale synthetic dataset, MOTSynth, and evaluated on several cross-domain benchmarks. It achieves competitive performance compared to various fine-tuning methods, as measured by standard accuracy metrics including minADE and minFDE.

**Questions:**

In addition to the concerns raised in the weaknesses section, I have the following additional questions:

1. What is the FPS of datasets? What is the actual time in seconds for the 9-step conditioned past and the 12-step predicted future?

2. Would it be possible to include an ablation on the number of K to understand how prediction accuracy scales with smaller/larger sample sizes?

3. How does the model perform when conditioned on fewer past timesteps than the default 9? Specifically, do the STES and PG-ES modules remain effective under shorter observation windows?

4. There appears to be a repeated citation of [48, 49].

**Ethical Concerns:**

["NO or VERY MINOR ethics concerns only"]

**Final Justification:**

Author's rebuttal has addressed my concerns regarding pool quality and model generalization. I will keep my score at borderline accept.

**Limitations:**

There is a lack of discussion around failure cases and limitations. For example, the paper could examine how performance degrades when STES selects suboptimal examples, or when the in-context pool has limited coverage.

**Paper Formatting Concerns:**

No major formatting issues observed.

**Quality:**

3

**Strengths And Weaknesses:**

Strengths:

1. The paper provides a range of experiments and ablation studies across multiple datasets, which helps assess the effectiveness of the proposed approach.

2. Figures included in the paper help illustrate key concepts and contribute to clarifying the proposed methodology.

3. The motivation for applying in-context learning to trajectory prediction is clear, and the challenges involved are well articulated and addressed in the introduction and related work sections.



Weaknesses:

1. My major concern is that while the paper discusses the size of the in-context example pool, it lacks sufficient analysis on the quality of the pool. The model’s performance could be sensitive to the diversity and representativeness of the examples, especially given the inherently stochastic nature of human trajectories. Subtle variations in motion—such as differences in speed or turning angles—can significantly affect outcomes. However, the paper primarily focuses on pool size, without examining how variation in pool quality impacts performance. This limits our understanding of how well the method generalizes to out-of-distribution behaviors or less frequent motion patterns.

2. Another concern is whether the motion patterns learned from the synthetic dataset (MOTSynth) adequately capture the diversity and complexity of real-world human behavior. It remains unclear how well the synthetic data reflects the intricate and multimodal nature of pedestrian motion. If the training data lacks this level of realism, it raises questions about the model’s ability to generalize beyond what it has seen during training. The paper would be strengthened by a discussion of which components of the framework, if any, contribute to such generalization on domain-specific datasets, and by providing more analysis or evidence regarding the realism and coverage of the training dataset.

3. The method relies on Social-Transmotion as the underlying architecture, but its generalizability to other trajectory prediction models is not evaluated. This raises questions about how tightly TrajICL is coupled with this specific backbone and whether the proposed in-context learning framework would transfer effectively to alternative architectures.

4. Since the paper emphasizes inference-time adaptability, it would be valuable to report inference time, particularly considering the cost of retrieving and encoding multiple in-context examples.

---

> ### Author Rebuttal · Authors · 2025-07-31
>
> Dear Reviewer od3z,
>
> We sincerely appreciate your insightful feedback. We have addressed the concerns raised in the review as detailed below.
>
> **Weakness 1: Effect of the Pool Quality on Prediction Accuracy**
>
> To analyze the effectiveness of the in-context pool quality, we examined the relationship between pool quality and prediction accuracy. We measured pool quality using the STES similarity metric. If the maximum similarity between a target trajectory and the trajectories in the pool is low, it suggests that the pool lacks similar examples.
>
> As we cannot upload figures, we categorized target trajectories into three similarity levels (low, medium, high) and summarized the results in the table below (experiments on MOTSynth, ADE reported).
>
> | Model | low | medium | high |
> |---|---|---|---|
> | Social-Transmotion | 37.0 | 12.1 | 5.45 |
> | TrajICL | **30.3** | **10.0** | **4.69** |
>
> This table indicates that when the target trajectory represents out-of-distribution behavior (e.g., unusual motion patterns), the performance is lower compared to in-distribution data. However, we would like to emphasize that even with such OoD trajectory inputs, our framework shows a significant performance improvement by using in-context examples (37.0 → 30.3).
>
> **Weakness 2: Generalization to real-world data**
>
> Thank you for your insightful feedback. While our synthetic training data may have limited motion diversity, our model generalizes effectively to real-world datasets like WildTrack, JRDB, and SDD.
>
> This strong generalization is primarily due to our PG-STES module. During inference, instead of relying solely on the synthetically trained model, PG-STES dynamically selects examples relevant to the target trajectory from a pool of test data. This process provides the model with in-domain knowledge of the current scene, effectively bridging the synthetic-to-real domain gap.
>
> This is partially verified in Table 3, where using randomly selected in-context samples does not improve performance on the JTA dataset.
>
> Furthermore, we verified this by measuring the Fréchet Inception Distance (FID) between the feature distributions of in-context examples on real WildTrack test data and the synthetic training dataset. Our PG-STES method achieves an FID score of 612, whereas a random selection from the same pool results in a vastly higher score of 2,454,212. This dramatically lower FID score demonstrates that our selection strategy is highly effective. By providing the model with the most pertinent real-world examples at inference time, PG-STES enables robust generalization, overcoming the limitations of the initial synthetic training.
>
> **Weakness 3: Generalization Capability with Different Backbones**
>
> As mentioned, we used Social-Transmotion as our backbone, as it is a state-of-the-art model with a generalized architecture. Our TrajICL framework is plugged in to enable in-context learning capabilities. To confirm its generalization, we replaced the backbone with ForecastMAE [1] (disabling the lane encoder for pedestrian prediction). The experimental results are summarized below:
>
> | Model | MOTSynth | WildTrack | JTA|
> | ----  | ---- | ----  | ---- |
> | ForecastMAE | 18.41/24.51  | 25.0/29.5 |0.76/1.23 |
> | ForecastMAE + TrajICL (4 shots) | **15.98**/**17.91**  |  **21.9/28.3**  | **0.64**/**0.91** |
>
> These results verify that our framework is generalized and can be applied to different backbones.
>
> [1] Forecast-MAE: Self-supervised Pre-training for Motion Forecasting with Masked Autoencoders, In ICCV 2023.
>
> **Weakness 4: Inference time**
>
> First, we would like to emphasize that our main motivation is to eliminate the fine-tuning and backpropagation costs on edge devices since the backpropagation process requires much higher GPU memory compared to the inference, as well as the burden of model management and environment-specific data collection (L.4, L.31-34, L.41...). This is what we refer to as "efficiency." We did not aim to accelerate inference time in the original paper.
>
> We will revise the term "efficiency" in the final manuscript to avoid confusion.
> However, as you pointed out, it is important to analyze the inference cost and compare it with the base model.
>
> | Model | FLOPs (GFLOPS) | Params (million) | Total Inference time (ms) |
> | ----  | ---- | ----  | ---- |
> | Social-Transmotion| 3.48  | 3.10 |6.11 |
> | TrajICL | 4.54  | 4.15  |7.69 + 0.04 (retrieval) |
>
> (Inference time measure with PC equipped with NVIDIA TITAN RTX GPU, 188 GB RAM, and Intel(R) Xeon(R) Silver 4216 CPU.)
>
> We follow the experimental setup of existing in-context learning approaches [1], where features for in-context samples are pre-computed. Our overall inference cost is higher because of the additional Predictor module for aggregating in-context sample features and the two-stage inference via PG-ES.
>
> However, since our primary motivation is to eliminate the training step after deployment, we also compared the fine-tuning cost of the base model with our inference cost:
>
> | Model | FLOPs (GFLOPS) |  GPU Memory (MB) |
> | ----  | ----  | ---- |
> | Social-Transmotion Fine-tuning | 10.43 | 462 |
> | TrajICL | 4.54  |288 |
>
> (GPU memory is computed with batch size = 1)
>
> Our required GPU memory is significantly less than that needed for fine-tuning the base model, which aligns with our paper's primary motivation. As mentioned in the conclusion, reducing the inference cost is a current limitation for edge-device deployment. We believe future work could explore applying existing techniques, such as Token Merging [2], to improve inference speed without requiring architectural modifications.
>
> [1] In-Context Learning Makes an Ego-evolving Scene Text Recognizer, In CVPR 2024.
>
> [2] Token Merging: Your ViT But Faster, In ICLR 2023.
>
> **Question 1: FPS of datasets**
>
> For MOTSynth and WildTrack datasets, the FPS is 2.0. The 9-step conditioned past corresponds to 4.5 seconds, and the 12-step predicted future corresponds to 6.0 seconds.
>
> For JRDB, JTA, and SDD datasets, the FPS is 2.5. The 9-step conditioned past corresponds to 3.6 seconds, and the 12-step predicted future corresponds to 4.8 seconds. We will incorporate this information into the final manuscript.
>
> **Question 2: ablation study of sample size**
>
> We would like to clarify that K denotes the number of predicted trajectories for the multimodal prediction. M controls the number of in-context samples. The ablation study for M is already conducted in Figure 3. The x-axis represents the number of input in-context samples (M), and the y-axis shows the accuracy on two benchmarks (MOTSynth and JTA). It demonstrates that increasing the number of samples improves performance.
>
> **Question 3: Fewer Timesteps evaluation**
>
> We conducted an additional experiment with fewer past timesteps during evaluation.
>
> Specifically, we reduced the input past timesteps from 9 to 6 and 3.
>
> | Model | 9 steps | 6 steps  | 3 steps |
> | ----  | ---- | ---- | ---- |
> | Social-Transmotion | 17.6/23.0  | 20.1/27.4 | 24.3/34.1 |
> | Ours w/ Random Selection | 16.6/21.8  | 19.0/23.6 | 19.3/25.7 |
> | Ours w/ STES Selection | 15.6/19.2  | 16.9/20.7 | 17.9/22.9 |
> | Ours w/ PG-ES Selection | **15.3**/**17.5**  | **16.0**/**18.0** | **16.4**/**19.3** |
>
> The table verifies that while performance degrades with fewer timesteps, the proposed selection strategies (especially PG-ES) still show significant performance improvement. For example, with 3 steps, PG-ES improves performance from 25.7 (Random) to 19.3, demonstrating the effectiveness of our approach even with limited input data.
>
> **Question 4: Repeated citation**
>
> Thank you for pointing out the duplicated citation. We will correct this in the final manuscript.
>
> **Limitation 1: Failure cases and Limitation**
>
> First, we would like to emphasize that we have already conducted experiments on scenarios with limited pool coverage (the few-shot scenarios in Table 2).
>
> We conducted additional experiments to analyze the effectiveness of PG-ES. Specifically, we measured the percentage of performance improvement from using STES to PG-ES based on the initial prediction error (ADE). As we cannot submit figures, we categorized the "noisiness" into three categories (low, medium, high) and summarized the results:
>
> |  | Low | Medium | High |
> | ----  | ---- | ---- |---- |
> | &Delta;	 |  -0.5% | -8.9% | -12.0% |
>
> A lower &Delta; value indicates a greater performance improvement from STES to PG-ES (lower ADE is better). The results confirm that PG-ES is specifically effective when the initial prediction is noisy.
>
> One important limitation is the inference time, as noted in the last section of our conclusion (L.321). Furthermore, improving robustness to noise in trajectories is another current limitation (see Reviewer 8wc2's response to W3) and a potential future research direction.

---

> > ### Comment · Reviewer_od3z · 2025-08-05
> > **Thank you for the rebuttal**
> >
> > Thank you for the thorough rebuttal. It has addressed my concerns, including the ablation on pool quality and the discussion on model generalization. I will keep my score and leave it to further discussion with other reviewers and AC.

---

> > > ### Author Response · Authors · 2025-08-06
> > >
> > > Dear reviewer od3z,
> > >
> > > We sincerely appreciate the reviewer for taking the time to provide constructive feedback.
> > > We will incorporate all the comments in the final manuscript.
> > >
> > > Best regards,
> > >
> > > Authors

---

### Official Review · Reviewer_eEf8 · 2025-07-02

**Clarity:** 4
**Significance:** 3
**Originality:** 2
**Rating:** 4
**Confidence:** 3

**Summary:**

The paper proposes TrajICL, an in-context learning (ICL) framework for pedestrian trajectory prediction that eliminates fine-tuning. Key innovations include: 1. Spatio-temporal similarity-based example selection (STES) for relevant context retrieval. 2. Prediction-guided example selection (PG-ES) leveraging predicted future trajectories to refine examples. 3. Training on large-scale synthetic data (MOTSynth) to enhance generalization. Experiments across 5 benchmarks (MOTSynth, JRDB, WildTrack, SDD, JTA) show TrajICL outperforms fine-tuned baselines in cross-domain adaptation.

**Questions:**

1.	The number of baseline methods compared by the authors is relatively limited. As the authors mentioned in the related work section, there are already some existing works on Adapting Trajectory Prediction. Although these methods have certain limitations in terms of computational costs and adaptability, I believe performance comparison experiments are still necessary. I also suggest that the authors include comparisons with relevant approaches such as Domain Generalization (DG) to further demonstrate the effectiveness of ICL. Additionally, since the authors mention some existing ICL methods in computer vision, performance comparisons with those should also be conducted.
2.	The paper does not seem to provide details or sensitivity analysis regarding the choice of the hyperparameter M. In addition, a runtime analysis should also be included in the paper.
3.	I’m also curious whether applying PG-ES multiple times would lead to further performance improvements?
4.	The paper lacks a bit in terms of originality. The authors should articulate the novelty and contribution of these operations within the proposed framework.

**Ethical Concerns:**

["NO or VERY MINOR ethics concerns only"]

**Final Justification:**

After reading the rebuttal and discussion, I am raising my score to Weak Accept (WA).

**Limitations:**

Yes

**Quality:**

3

**Strengths And Weaknesses:**

Strengths:

Concise and Accessible Writing: This paper is easy to read and follow.
Practical Contribution: Addresses edge-device constraints by replacing fine-tuning with lightweight ICL.
Rigorous Evaluation: Extensive cross-dataset validation (5 benchmarks) with consistent improvements.
Limitations Discussed: Notes computational trade-offs of increasing in-context examples.

Weaknesses:

The experimental analysis is not sufficiently thorough.
The paper need more baseline comparison.

---

> ### Author Rebuttal · Authors · 2025-07-31
>
> Dear Reviewer  eEf8,
>
> We sincerely appreciate for providing the insightful feedback. We have addressed the concerns raised in the review as below.
>
> **Weakness 1, Question 1.1: Comparison with adaptive trajectory prediction methods**
>
> Thank you for this constructive comment. In most existing adaptive trajectory prediction methods, the adaptation module and the prediction model architecture are tightly coupled [1,2,3]. This, along with differences in experimental settings (e.g., training datasets, target domains), makes a fair, "apple-to-apple" comparison difficult. These previous works also compared with baselines or commonly-used trajectory prediction models.
>
> We have already included a comparison with the conceptual methodology from a recent adaptation paper [1] in our main experiments (Table 1, "+VPT Shallow"). We adapted their visual prompt tuning approach to fit within our baeline model.
>
> For further context, we can review the performance of our approach against recent adaptive methods [2,3] on the same evaluation datasets, as summarized below. We have also included experiments on the NBA SportsVU dataset [4].
>
> | Model  | SDD | NBA |
> | ----  | ---- | ---- |
> | [2] |  11.0/20.0 |  1.28/2.64 |
> | [3] |  10.1/17.1 |  ---/--- |
> | Ours | 8.40/14.8 | 0.92/1.37 |
>
>
> This table shows that our approach outperforms these recent adaptive methods. However, it is important to note that differences in the base model architecture and training datasets exist between our work and theirs.
>
> [1] Adaptive Human Trajectory Prediction via Latent Corridors. In ECCV 2024.
>
> [2] Recurrent Aligned Network for Generalized Pedestrian Trajectory Prediction. In IEEE TCSVT 2024.
>
> [3] MetaTra: Meta-Learning for Generalized Trajectory Prediction in Unseen Domain, In arXiv 2024.
>
> [4] Learning fine-grained spatial models for dynamic sports play prediction, in IEEE ICDM 2014.
>
> **Question 1.2: Comparison with ICL methods in computer vision**
>
> Since the tasks, loss functions, and model designs of other In-Context Learning (ICL) approaches in computer vision are completely different from ours, a fair and direct comparison with these works is not possible.
>
> However, we can adapt their proposed in-context sample selection strategies to our trajectory prediction task. The experimental results on the MOTSynth dataset are summarized below:
>
> | Selection Method | Paper References | ADE| ADE |
> | ----  | ---- | ---- | ---- |
> | Random  | [1, 2]  | 16.6 |  21.8   |
> | Feature Similarity  | [3,4]  | 16.0 |  19.2   |
> | Motion Similarity   | [5] | 19.2  |  32.5   |
> | PG-ES (Ours) | --- | **15.3** |  **17.5** |
>
> These results verify that our selection strategy, PG-ES, achieves the best performance on the trajectory prediction task.
>
> [1] AIM: Let Any Multimodal Large Language Models Embrace Efficient In-Context Learning, In AAAI 2025.
>
> [2] MMICL: Empowering Vision-Language Model with Multi-modal In-Context Learning, In ICLR 2024,
>
> [3] Flamingo: a Visual Language Model for Few-Shot Learning, In NeurIPS 2022.
>
> [4] VideoICL: Confidence-based Iterative In-context Learning for Out-of-Distribution Video Understanding, In CVPR 2025.
>
> [5] Skeleton-in-Context: Unified Skeleton Sequence Modeling with In-Context Learning, In CVPR 2024.
>
> **Question 2: Effect of hyperparameter M, the number of in-context samples**
>
> We would like to note that the ablation study for M, the number of in-context samples, is already included in Figure 3 of the paper. The x-axis shows the number of input in-context samples (M), and the y-axis shows the accuracy on two benchmarks (MOTSynth and JTA). The figure clearly demonstrates that increasing the number of samples improves performance.
>
> **Question 3: Performance Improvement via Iterative Application of PG-ES**
>
> Thank you for this insightful comment. We conducted an additional experiment applying PG-ES multiple times. The results on the MOTSynth dataset are summarized below:
>
> | Repetition | FDE |
> | ----  | ---- |
> | 1x  |17.5   |
> | 2x   | 16.6  |
> | 3x |  16.7 |
> | 4x |  16.7 |
>
> The results show that applying the method two times improves performance by 5.2% and then converges, with no further significant gains.
>
> **Question 4: Clarification of contributions**
>
> The adaptability of a trajectory prediction model is a critical challenge for real-world applications (as noted by Reviewer xGzb's Strength 1, Reviewer od3z's Strength 3, and Reviewer 8wc2's Strength 1). To address this, we propose an in-context learning (ICL) approach that, unlike existing methods, eliminates the need to update model weights in the target environment. This is inspired by the recent advancements in Large Language Models (LLMs). This is the first work to apply ICL to pedestrian trajectory prediction, and the connection between the challenge of adaptability and the conceptual idea of LLM's ICL is a novel perspective to the field (as noted by Reviewer 8wc2's Strength 2).
>
> We would like to re-emphasize our contributions as follows:
>
> **Effective Sampling Strategy**: As shown in Figure 3, a naive approach using randomly selected samples does not improve performance. We believe this is due to the inherent complexity of trajectory data. Our proposed sampling strategy, PG-ES, is based on spatio-temporal similarity in the input space and a two-round inference process. These ideas for sampling are novel within the context of other ICL approaches in computer vision and demonstrate clear effectiveness in trajectory prediction tasks (as shown in our responses above).
>
> **Extensive Experiments**: Existing adaptability approaches have typically been evaluated on at most three datasets (ETH-UCY, SDD, NBA) with similar data setups (e.g., surveillance cameras, pedestrian motion). In contrast, we have verified the effectiveness of our framework using seven datasets: MOTSynth, JRDB-Image, WildTrack, SDD, JRDB-World, JTA, and NBA SportsVU. This broad evaluation demonstrates the wider applicability of our framework to various domains.
>
> We will emphasize the novelty and contributions of our work in the final manuscript.

---

### Official Review · Reviewer_xGzb · 2025-07-02

**Clarity:** 2
**Significance:** 3
**Originality:** 3
**Rating:** 4
**Confidence:** 4

**Summary:**

This paper presents an In-Context Learning framework for pedestrian trajectory prediction.  To address adaptability of prediction in different domains, they introduce a method without fine-tuning. A spatio-temporal similarity-based example selection and a further refine strategy are proposed. They also use synthetic dataset for training to simulate diverse outdoor environments.
Experiments on multiple datasets are conducted to verify its effectiveness. Both in-domain and cross-domain settings are evaluated.

**Questions:**

1)	The computational complexity and efficiency are not discussed.

2)	Some implementation details to be complemented;

3)	More discussions or explanations on the few-shot comparison;

4)	Dependence of PG-ES on the quality of the trajectory predictor in the first step need be analyzed.

**Ethical Concerns:**

["NO or VERY MINOR ethics concerns only"]

**Final Justification:**

THe rebuttal has solved my concerns. I keep my rating Borderline accept.

**Limitations:**

The limitation analysis is not sufficient. The computational complexity and efficiency are not discussed. The comparison with other prediction models considering adaptability seems incomplete.

**Quality:**

3

**Strengths And Weaknesses:**

Strength
1）	It states clearly on motivation and problems of existing methods.

2）	The idea of using In-Context Learning for adaptability is interesting in pedestrian trajectory prediction.

3）	Ablation studies on the two designed modules are provided, i.e., similarity-based example selection module and prediction-guided refinement.

4）	Figures are clearly displayed including visualization of qualitative results.

Weakness:

--In Equa(3), K appears for the first time and k is undefined. It can be declared at the beginning such as in the Problem Formulation section that the study focuses on multi-modal or stochastic prediction.

Unclear implementation details
--in line 194-195 “PredictionHead, which consists of K simple one-layer MLPs for multimodal prediction”. In line 244 “Predictor consists of three layers and four attention”. These two statements are inconsistent. What’s the exact prediction head’s configuration? Why use four attention modules? Maybe it can be added in supplementary material.

--Different setting:
It comprises two sequential training phases—vanilla trajectory prediction training and in-context training. Most existing trajectory prediction methods do not use the two-stage paradigm. Though some methods consider adaptability，they adopt one-stage framework.  It is said the proposed TrajICL “offers greater efficiency”. The computational efficiency is suggested to be reported and compared.

-- From Table2, when example pool size decreases from 100% (in Table 1) to 10% and 20%, prediction performances degenerate differently on different datasets. The degeneration percentage (not reported in Table 2) is greater on SDD.   I wonder its few-shot performance on the more challenging NBA SportVU Dataset. Whether TrajICL is affected by the extent of distribution divergence between training and inferring datasets.


--it is unclear whether other trajectory prediction methods considering adaptability through transfer learning, online adaption, et al. are comparable.


--The number of trainable parameters is suggested to be compared in Table 1, so it can better support the claim TrajICL is lighter and offers greater efficiency.

--The PG-ES consists of two steps. Its effect will be dependent on the quality of the trajectory predictor in the first step, since future trajectories in the second step are produced by the predictor in the first step. From Table3, PG-ES improves over STES, whose improvement is greater than those from spatial and temporal components in the STES. What if low-quality or noisy future trajectories are produced by the trajectory predictor in the first step?

--From Table4, VTP training has tiny improvement over without VTP training. That means, the original trajectory prediction model without VTP training is strong enough. What if the In-Context-Learning works on a weaker base?

---

> ### Author Rebuttal · Authors · 2025-07-31
>
> Dear Reviewer xGzb,
>
> We sincerely appreciate for providing the insightful feedback. We have addressed the concerns raised in the review as below.
>
> **Weakness 1.1, Question 2: Clarification of K.**
> Thank you for pointing out the undefined variable. We will add an explanation for k (number of predictions) in the problem formulation section of the final manuscript.
>
> **Weakness 1.2, Question 2: Difference between PredictionHead and Predictor.**
> We apologize for the confusion. PredictionHead and Predictor are distinct modules, as illustrated in Figure 2. The PredictionHead consists of K one-layer MLPs, while the Predictor is composed of a three-layer Transformer Encoder with four attention heads. The Multimodal Decoder in Figure 2 is the PredictionHead. We will revise the manuscript to clarify this distinction for readers.
>
>
> **Weakness 2, Weakness 6, Limitation 1, Question 3:Different Setting, Clarification of efficiency.**
> We would like to clarify that the "efficiency" discussed in the paper refers to the inference phase, not the training phase. For example, our method eliminates the need for backpropagation to fine-tune the model on target domain data, resulting to reduce GPU memory after the deployment by utilizing the in-context samples.
>
> As noted, our two-stage training paradigm does increase the overall training cost (e.g., GPU hours) compared to the base model. However, a key advantage of our approach is that after training on the synthetic MOTSynth dataset, the model does not require any additional data collection or training on the target environment.
>
> For reference, the number of trainable parameters for Social-Transmotion is 3.1 million, while TrajICL has 4.15 million.
>
>
> **Weakness 3, Question 3: Few-shot performance on the more challenging NBA SportVU Dataset.**
> Thank you for this insightful comment. We have evaluated our approach on the NBA SportVU dataset (Rebounding and Scoring subsets) using different pool sizes. The results are shown in the table below:
>
> | Model  | Rebounding Subset	 | Scoring Subset |
> | ----  | ---- | ---- |
> | Social-Transmotion |  1.03/1.57 |  1.02/1.91 |
> | TrajICL (10%) |  0.93/1.61 |  0.98/1.78 |
> | TrajICL (20%) | 0.92/1.58 | 0.97/1.75 |
> | TrajICL (100%) | **0.90**/**1.51** | **0.95**/**1.72** |
>
> (X%) denotes the pool sizes.
>
> This table indicates that our approach performs well even in challenging scenarios with a large distribution divergence between the training and testing datasets.
>
> Interestingly, the performance drop from a 100% to a 10% pool size is relatively small (+4% degradation). This may be because, despite the high complexity of human motion in the NBA SportVU dataset, the target motions might be similar to those in the in-context pool. This allows the model to effectively use the pool information for trajectory prediction. In contrast, if a dataset like SDD has a wide variety of motions, it might be more difficult to leverage in-context information effectively during inference.
>
> **Weakness 4, Limitation 1: Comparison with adaptive trajectory prediction methods**
>
> Thank you for your constructive feedback. In most existing adaptive trajectory prediction methods, the adaptation module and the prediction model architecture are tightly coupled [1,2,3]. Furthermore, they often use different training settings (e.g., training datasets and target domains), making a fair, "apple-to-apple" comparison difficult. These previous works also compared with baselines or commonly-used trajectory prediction models.
>
> We have already included a comparison with the conceptual methodology from a recent adaptation paper [1] in our main experiments (Table 1, “+VPT Shallow”). This involved adapting their visual prompt tuning approach into our base model (Social-Transmotion).
>
> To provide further context, we can review the performance of recent adaptive methods [2,3] on the same evaluation datasets, as summarized below:
>
> | Model  | SDD | NBA |
> | ----  | ---- | ---- |
> | [2] |  11.0/20.0 |  1.28/2.64 |
> | [3] |  10.1/17.1 |  ---/--- |
> | Ours | **8.40**/**14.8** | **0.92**/**1.37** |
>
> This table shows that our approach outperforms these recent adaptive methods. However, it is important to note that significant differences in base model architecture and training datasets exist between our work and theirs.
>
> [1] Adaptive Human Trajectory Prediction via Latent Corridors. In ECCV 2024.
>
> [2]  Recurrent Aligned Network for Generalized Pedestrian Trajectory Prediction. In IEEE ITCSVT 2024.
>
> [3] MetaTra: Meta-Learning for Generalized Trajectory Prediction in Unseen Domain, In arXiv 2024.
>
> **Weakness 6, Question 4: Effect of the quality of the trajectory predictor in the first step for PG-ES**
>
> Thank you for this insightful comment. We conducted an additional experiment to analyze the effectiveness of PG-ES. Specifically, we measured the percentage of performance improvement from using STES to PG-ES based on the initial prediction error (ADE).
>
> Since we cannot submit figures in this rebuttal, we have categorized the initial "noisiness (= ADE at the first inference)" into three (low, medium, high) and summarized the results below:
>
> |  | Low | Medium | High |
> | ----  | ---- | ---- |---- |
> | &Delta;	 |  -0.5% | -8.9% | -12.0% |
>
> A lower &Delta; value indicates a greater performance improvement from STES to PG-ES (lower ADE is better). The results confirm that PG-ES is particularly effective when the initial prediction is noisy (High means the initial prediction is distant from the ground-truth).
>
> PG-ES offers the greatest benefit for trajectories with high initial errors because these are typically cases where a similar past could lead to multiple ambiguous futures. By generating several future hypotheses and re-selecting examples that match one of them, the model effectively corrects the initial poor selection. This correction process leads to the most significant improvement where the initial prediction was wrong.
>
> **Weakness 7: Performance of  weaker base model**
>
> We would like to clarify the training setup of our approach.
>
> In the first VTP training stage, the encoder is trained to predict the trajectory of a target pedestrian from target and other pedestrians' trajectories. In the second in-context training stage, all modules—the encoder, predictor, and prediction head—are trained to predict the trajectory using both the target and in-context samples. Therefore, even without VTP training, the model can acquire in-context learning capabilities by backpropagating the loss through the entire network.
>
> As you suggested, we trained a weaker base model with a reduced feature dimension size (from 128 to 64) without VTP on MOTSynth. Due to the limited time for this rebuttal, we experimented with a 4-shot model.
>
> | | ADE| FDE |
> | ----  | ---- | ---- |
> | zero-shot |  16.41 | 19.36 |
> | 4-shot |  **15.30** | **17.50** |
>
> This table shows that the in-context capability is still acquired with a weaker base model.

---

> > ### Comment · Reviewer_xGzb · 2025-08-05
> >
> > I appreciate the detailed response, including complemented experiments, comparison with other adaptive methods, ablation study on the initial prediction, ablation on the base model quality and explainations of settings. THese have solved my concerns.

---

> > > ### Author Response · Authors · 2025-08-05
> > >
> > > Dear reviewer xGzb,
> > >
> > > We sincerely appreciate for reading our rebuttal. We are glad to hear that the concerns have been resolved.
> > > Based on your feedback, we will revise the final manuscript accordingly:
> > >
> > > - Clarify the notation, model architecture, and "efficiency".
> > > - Include the experiments with other adaptive trajectory prediction approaches.
> > > - Include the detailed analysis of PG-ES and the weaker base model.
> > >
> > > Once again, thank you for your thoughtful feedback and for taking the time to help improve our work.
> > >
> > > Best regards,

---

### Comment · Area_Chair_ESv9 · 2025-08-03
**Discussion**

Dear reviewers,

Thank you for your hard work. We have now entered the Author-Reviewer Discussions phase (July 31 - August 8). Could you please read all the reviewers' comments and the authors' responses as soon as possible, and provide your first feedback? For example, you may indicate whether the authors have answered and addressed your questions.

Thank you!
AC

---

### Decision · Program_Chairs · 2025-09-17

**Decision:**

Accept (poster)

**Comment:**

This work presents an In-Context Learning framework for pedestrian trajectory prediction. All reviewers were positive about this work, e.g., though this work is clearly on motivation, the idea of applying ICL to trajectory prediction is novel, and the results are strong and promising, and easy to follow. Moreover, their concerns were addressed in the rebuttal and discussion.

Overall, I recommend accepting this paper to NeurIPS.